# Aerogels Part 1: A Focus on the Most Patented Ultralight, Highly Porous Inorganic Networks and the Plethora of Their Advanced Applications

**DOI:** 10.3390/gels11090718

**Published:** 2025-09-08

**Authors:** Silvana Alfei

**Affiliations:** Department of Pharmacy (DIFAR), University of Genoa, Viale Cembrano, 4, 16148 Genoa, Italy; alfei@difar.unige.it; Tel.: +39-010-355-2296

**Keywords:** aerogels (AGs), sol–gel method, inorganic-based AGs, silica oxide AGs, metal oxide AGs, alumina oxide Ads, zirconia oxide AGs, titania oxide AGs

## Abstract

Aerogels (AGs) are highly porous, low-density, disordered, ultralight macroscopic materials with immense surface areas. Traditionally synthesized using aqueous sol–gel chemistry, starting by molecular precursors, the nanoparticles (NPs) dispersions gelation method is nowadays the most used procedure to obtain AGs with improved crystallinity and broader structural, morphological and compositional complexity. The Sol–gel process consists of preparing a solution by hydrolysis of different precursors, followed by gelation, ageing and a drying phase, via supercritical, freeze-drying or ambient evaporation. AGs can be classified based on various factors, such as appearance, synthetic methods, chemical origin, drying methods, microstructure, etc. Due to their nonpareil characteristics, AGs are completely different from common NPs, thus covering different and more extensive applications. AGs can be applied in supercapacitors, acoustic devices, drug delivery, thermal insulation, catalysis, electrocatalysis, gas absorption, gas separation, organic and inorganic xenobiotics removal from water and air and radionucleotides management. This review provides first an analysis on AGs according to data found in CAS Content Collection. Then, an AGs’ classification based on the chemical origin of their precursors, as well as the different methods existing to prepare AGs and the current optimization strategies are discussed. Following, focusing on AGs of inorganic origin, silica and metal oxide-based AGs are reviewed, deeply discussing their properties, specific synthesis and possible uses. These classes were chosen based on the evidence that they are the most experimented, patented and marketed AGs. Several related case studies are reported, some of which have been presented in reader-friendly tables and discussed.

## 1. Introduction

Aerogels (AGs) are porous, ultra-lightweight, nanostructured materials, commonly obtained from a variety of hydrogels by substituting the liquid component with a gas, using different drying techniques. The first AG was synthesized starting from silica precursors by Kistler in 1931 [1]. By the 1990s, NASA used AGs for thermal insulation in spacecraft, space suits, and blankets. Over the years, AGs have been adopted for insulation in subsea systems, oil refineries, industrial pipelines, buildings, refrigerators, and clothing like jackets and shoe inserts [2]. The synthetic methods mostly used to prepare AGs first provide hydrogels, which need drying methods to become AGs. Concerning this, the most adopted techniques used to transform hydrogels into AGs strongly contribute to the physical characteristics of the resulting materials [3]. The use of advanced drying procedures, such as the supercritical CO_2_ method, can lead to the formation of a robust, ultra-lightweight, dendritic microstructure consisting of pores smaller than 100 nm and 90 to 99.8% empty space [4]. Such tiny pores are too small for air to travel through, thus making AGs highly efficient as insulation materials [2]. AGs display high specific surface area (SSA), a low mean free path for diffusion, low thermal conductivity, low acoustic velocity, low refractive index, low dielectric constant and extremely low density, ranging from 0.0011 to ~0.5 g/cm^3^ [5]. A large variety of AGs have been produced, which demonstrated to be promising candidates for other applications different from insulation, which include their use as catalysts [6,7], supports for catalysts [8], sensors [9], filters [10], cosmic dust collectors [11,12], detectors in particle physics [13,14], thermal insulators [15], interlayer dielectrics, optical applications [9] and many others [9,16,17,18,19]. Additionally, as conductive matrices, AGs could be potentially applied in battery materials, capacitors and components in fuel- or solar cells [20]. Anyway, despite their huge application potential, the sol–gel process, which is the most popular method to synthesized AGs, is not completely controllable and requires costly precursors [21], thus limiting their wide translation in daily practice. Moreover, hydrogels obtained via the sol–gel route are for the most part amorphous and need calcination at high temperatures for crystallization, which can lead to the loss of many properties typical for AGs. In fact, particle growth and coalescence lower the specific surface area (SSA) and the porosity of AGs, leading to the collapse of the structure and destruction of the monolithic body. Recent advancements have allowed scientists to prepare more durable materials with enhanced structural integrity and thermal properties. Several types of AGs are known, including inorganic, organic and composite AGs. Among inorganic AGs, those derived by metal oxide precursor, especially when they are mixed oxides, are hard to be synthesized in gel form, since it is difficult to adjust and control the hydrolysis of the molecular precursors and condensation rates, which are the first steps of the sol–gel method [22]. However, versatile sol–gel routes, including the dispersed inorganic sol–gel (DIS) method and epoxide addition (EA) process, which use inorganic salt solutions as reagents, polyacrylic acid (PAA) as template and propylene oxide (PO), have expanded the range of available materials [23]. By such approaches, several monolithic AGs have been achieved, but the optimal crystallinity has not yet been reached [23]. Furthermore, pure metallic AGs are generally not directly synthesized via molecular routes and need thermal treatments in reducing atmospheres [24]. Anyway, the recent use of preformed nanoparticles (NPs) as building blocks to be assembled for achieving AGs represents an elegant and likewise powerful method to overcome the abovementioned issues [25]. Nevertheless, the use of templates is needed to control the assembly behaviour of NPs into a 3D percolating network, which could be stable enough to allow further processing.

### 1.1. Current Research Hotspots and Future Development Trends

Collectively, in our opinion, we can summarize that the current main research hotspots of AGs focus mainly on their synthesis, mechanical capacity regulation, and their applications for thermal insulation, adsorption, and Cherenkov detector radiators. Specifically, sol–gel process, as a kind of synthesis method of AGs, is the most popular research topic, followed by the mechanical property regulation of AGs, which plays an important part in the process of machining and by their applications as thermal insulators, absorbents and to engineered Cherenkov detector radiators. Other minor hotspot applications follow, including the use of AGs as catalyst carriers, drug delivery, and optical devices. However, despite the extensive study of AGs, there are still many challenges and opportunities in the future for these materials. Future development trends should focus on eco-friendly synthesis, which is critical to environmental pollution and the health of operators. The preparation of AGs using green and pollution-free reagents, such as rice husk ash as a non-toxic silicon source, to produce silica AGs (SAGs) needs to be considered. Additionally, a synthetic method, using a non-toxic solvent in the process of preparation and ageing, needs to be further explored.

Although many studies focused on the enhancement of hardness and compressibility performance of AGs exist, other bearing conditions, such as twist and bend, should be discussed.

Despite the development of 3D printing allowing experts to manufacture structurally different products, computer techniques, such as simulations, should be applied to design novel structure of AGs and to simulate the total deformation ability of products with different structures. This could enable scientists to determine the structure with the best mechanical properties and deformation ability.

Although 3D printing technology can directly produce 3D printing objects from a computer-aided design model, thus endowing them with potential for mass manufacturing, 3D-printed objects need to be solidified to maintain an original shape after 3D printing. In this regard, research on solidification methods is still in its infancy and needs improvements, mainly in terms of thermal solidification and freeze casting. Additionally, combining freeze casting and a more intelligent temperature control system could become a scale production method.

### 1.2. An Analysis of AGs According to the CAS Content Collection

As abovementioned, based on the chemical composition of their precursors, AGs can be classified as inorganic, organic and composites AGs, whose subclasses have been disserted in detail later in this review [26]. An analysis of the CAS Content Collection has provided evidence of the extensive growth in publications, in terms of journals and patents, relating to AGs over the last two decades. Mainly from 2013, the number of patents has risen consistently, even more than journals in some years, indicating a significant commercial interest in this field (Figure 1).

Patents have even exceeded journals both for inorganic AGs that have been in the market for many years and for the newer synthetic polymer-based AGs, as evidenced in Figure 2.

Conversely, the very recent composite AGs have understandably fewer related publications, and journals are slightly outpacing patent publications at the moment. In the following Figure 3, the compounds most used as precursors to achieve particular classes of AGs have been reported, according to the CAS Content Collection™.

Despite the AGs market is still in its infancy, due to the survival of certain challenges not yet fully solved, the future for these materials is bright, and their marketplace is expected to experience an annual growth rate (AGR) of approximately 17% over the future period of 2025–2035. This is because AGs are showing considerable promise in energy storage, catalysis, and various biomedical applications, as well as cosmetics and acoustics (soundproofing) industries, as confirmed by the several patent publications [2]. In fact, patents surpassed publications in journals, thus establishing their translation from academic and laboratory settings to the industrial one and highlighting the high commercialization of AGs in the abovementioned fields (Figure 4).

Due to their ability to diminish the speed and amplitude of sound waves, AGs can be used as acoustic insulation materials, while in cosmetics, AGs can act as anti-shine agents in creams. Composite titanium and silica oxide AGs are being added to sunscreen because of their photoprotective properties, which can increase the sun protection factor (SPF) over that of conventional formulations.

In this promising scenario of AGs, the main purpose of this paper is to give an updated view on the current state of the art of AGs, both in academia and industry. The following sections provide first an AGs’ classification based on the chemical origin of their precursors. Then, the different methods which exist to prepare AGs, with related optimization strategies, are discussed. Specifically, the traditional sol–gel and not sol–gel chemical routes involving molecular precursors, up to the sol–gel methods using NPs as building blocks, are discussed. Following, focusing on AGs of inorganic origin, silica and metal oxide-based AGs are reviewed, deeply discussing their properties, specific synthesis and possible uses. These classes were chosen based on the evidence that they are the most experimented, patented and marketed AGs. Several related case studies have been reported and discussed, some of which have been presented in reader-friendly tables.

## 2. Classification of Aerogels According to Their Chemical Origin

A detailed classification of AGs based on the chemical origin of their precursors has been reported in Table 1.

Inorganic aerogels (IAGs) include silica AGs, chalcogenides, oxide-based, carbide-based, nitride-based, and metal/noble metal AGs [2]. IAGs are typically synthesized from inorganic precursor materials, like metal alkoxides or metal salts. In this class, silica aerogels (SAGs) possess a variety of nonpareil properties, such as low density (0.003–0.5 g/cm^3^), large porosity (80–99.8%), low thermal conductivity, allowed by their small mesopores (0.005–0.1 W/(mK)), ultra-low dielectric constant (*k* = 1.0–2.0) and a low refractive index (1.05) [54]. SAGs have technological applications, mainly based on their low thermal conductivity. In fact, the market for SAG thermal super insulators is growing quickly worldwide. Compared to other traditional insulation materials, such as fibreglass, foam plastic and polystyrene, SAGs excel in many areas where they fall short. Their superior thermal performance, energy efficiency, moisture resistance, and fire safety make them a top choice. While the initial investment may be higher, the long-term benefits cannot be overlooked. Collectively, SAGs outperform traditional insulation materials due to their higher R-value per inch and because they possess the lowest thermal conductivity among solid materials. Furthermore, it is also possible to consider applying SAGs as gas filters and dielectric materials, and/or for acoustic insulation and catalysis, even if they have not yet had a substantial impact on the market in these terms [55].

Chalcogenide-based AGs (CAGs) or metal chalcogenide AGs (MCAGs) are made from chalcogenides [28], which include sulphides, selenides, tellurides and polonides. MCAGs are preferentially applied to absorb heavy metals [56], such as mercury, lead and cadmium from water [56], while MCAGs of non-platinum metals are very good for desulfurization [57].

Additionally, high surface area MCAGs are promising for gas separation [58,59] and for capturing radionuclides from nuclear waste such as ^99^Tc, ^238^U, and especially^129^I [60]. In all these applications, MCAGs are superior to the traditionally used materials due to the covalent interaction with a smaller electronegativity difference and the facile polarization of chalcogen elements in metal–chalcogen bonding, and to the poor affinity for hard acidic alkali metal ions and their highly polarizable surface. Porous transition-metal dichalcogenides, such as molybdenum disulfide (MoS_2_) and tungsten disulfide (WS_2_) are electrocatalysts and photocatalysts are more efficient than traditional ones due to their tuneable bandgaps and large catalytically active surface areas, with no need for extra functionalization to enhance chemical activities. In addition, MCAGs can capture and sequester carbon dioxide (CO_2_) from the atmosphere, which helps in decreasing the amount of greenhouse gases released and mitigating the effects of climate change. Oxide-based aerogels (OAGs) are the most extensively used AGs. Various metal oxide AGs, including stannic oxide and tungstic oxide AGs, were prepared by Kistler et al. for the first time [32], but over the years, zirconia, titania, alumina, magnesium oxide and cobalt–nickel oxide AGs have been extensively experimented on [29,30,31]. In addition to the OAGs reported here and in Table 1, which are among the most studied ones, there are also examples for which the literature is much more limited [9]. They include Fe–Cr–Al mixed oxides, gold–iron oxides, Li_2_O·B_2_O_2_, GeO_2_, ZnO and V doped ZnO, Sn–Al oxides, Nb_2_O_5_, Pd doped CeO_2_, MnO_2_, Ta_2_O_5_ and more complex compositions like MgFe_2_O_4_, BaTiO_3_, SrTiO_3_, PbTiO_3_, Li_4_Ti_5_O_12_, VOHPO_4_·0.5H_2_O, La_2_Mo_2_O_9_ and CuO–Zr*_x_*Ce_1−*x*_O*_y_* [9].

Carbide-based AGs (CRAGs) include silicon carbide, carbide-derived carbons and molybdenum carbide AGs. Such materials have been extensively applied as catalysts for hydrogen production, electrodes in supercapacitors, gas turbine components, heat dissipation substrates and heat exchangers [33,34,35].

Nitride-based AGs (NAGs) comprise carbon nitride, aluminum nitride [36,37] and boron nitride AGs. The latter is mostly applied as a photocatalyst and to construct phase change materials from smart devices or used for pollutant degradation or removal [38,39].

Metal aerogels (MAGs) and noble metal aerogels (NMAGs) represent a singular class of inorganic AGs, which possess the properties of both metals and AGs and are widely applied in detection-based sectors [40,61]. MAGs and NMAGs are obtained from networks of metal NPs that have been super critically dried to produce nanofoams mainly of iron, gold and silver [40].

Organic aerogels (OAGs) include polysaccharide-based, phenolic-based, polyol-based, protein-based, and carbon-based materials. OAGs were researched because of the non-biocompatibility of most IAGs. Polysaccharide-based aerogels (PAGs) are obtained using natural saccharide polymers, like alginate, cellulose, pectin, starch, chitosan, carrageenan, mucilage, etc., which represent renewable resources. Methods to prepare PAGs involve dissolving polysaccharides in organic solvents or using crosslinking reactions or ionic liquids, followed by drying to remove liquid [41]. PAGs are currently applied for drug delivery, catalysis, environmental remediation or as thickeners [42].

Phenolic-based aerogels (PHAGs) were first developed by Pekala et al. using a condensation reaction between resorcinol with formaldehyde [45]. Nowadays, PHAGs are prepared via the same reaction using basic and acidic activators, where gelation occurs by phenolic furfural crosslinking [44]. PHAGs are mainly applied as reinforcement materials, for thermal and sound insulators [43].

Polyol based aerogels (POAGs), including polyurea AGs, are organic materials containing multiple hydroxyl groups, which are obtained via a polycondensation reaction between multifunctional isocyanates and amines, water or mineral acids. Due to their optical transparency, POAGs can be used in sensors and the glaring system of insulating windows [46,47].

Protein-based aerogels (PRAGs) are manufactured starting from proteins isolated from plants and animal sources like whey, albumin, collagen, gluten, silk fibroin, etc. PRAGs are biodegradable and biocompatible and are mainly finalized to tissue engineering as scaffold materials or are used in the food industry for encapsulating bioactive food constituents or for enhancing food properties like texture, taste and nutritional values [48,49,50].

Carbon-based aerogels (CBAGs) could belong to the class of IAGs, due to their carbonaceous nature. However, they are classified as organic-based materials due to the origin of biological sources, which are pyrolyzed or combusted to produce them. Moreover, they are considered organic materials since some of them are produced using organic precursor materials like phenol formaldehyde resin. CBAGs mainly include carbon nanotube, graphene and polymeric AGs [2]. CBAGs are mainly exploited as adsorbents, as well as electrodes in electrical devices [51,52,53].

Finally, composite aerogels (COAGs) include mixed-oxide AGs, aerogel–metal–organic framework (MOF) composites, aerogel–Mxene composites, and aerogels synthesized by combining inorganic and organic precursors [2]. Table 2 summarizes the key properties and main applications of the main types of AGs.

## 3. Synthetic Methods to Achieve Aerogels

### 3.1. Molecular Routes to Aerogels (AGs) by Wet Sol–Gel Processes

The traditional methods to prepare AGs start from molecular precursors of different types and use aqueous sol–gel chemistry, which is a useful technique for altering substrate surfaces and producing large surface areas and stable surfaces [62]. The sol–gel process basically involves the hydrolysis of precursors in acidic or basic solutions, followed by polycondensation of the hydrolysed products [63]. Ionic liquids are special solvents that are considered better solvents for numerous chemical processes, including AG preparation, due to their low vapour pressure and wide range of characteristics [63]. Figure 1 simplifies such a chemical sequence, in which the precursors are chosen as metal alkoxides to provide metal oxide AGs (MOAGs) [62].

By nucleation and growing progressions, the alkoxide hydrolysis followed by the condensation step forms an amorphous colloidal dispersion [9]. Further condensation and crosslinking reactions beyond the so-called sol-to-gel transition lead to an open-porous network (gel), which still includes a continuous liquid phase [64]. The main problematics related to the sol–gel procedure concern the highly rapid rate of the hydrolysis and condensation stages, which are difficult to control, thus rendering it hard to regulate the gelling conduct and to tune the porosity of the final material. Several strategies have been implemented to influence the reaction kinetics of the sol–gel process and to control morphology and particle size [64], including the use of precursors with bulky inert side chains, the variation in precursors concentrations and/or of the solvents, the use of surface-active agents, the adjustment of pH and many more [65]. Furthermore, kinetic control remains a major challenge in the production of AGs. Non-optimal hydrolytic conditions could have detrimental effects on gelling behaviour [66], which is the second substantial step towards the AG formation [32]. Moreover, once the gel network is formed, several unreacted groups can still survive in the gel backbone and further structural changes within the gels are needed to obtain a more mechanically stable scaffold. Enough time is necessary for the gel network to be strengthened by managing the overlaying solution’s pH, concentration, and water content. To this end, once formed, gels are often aged and ripened before supercritical, spry-, freeze- or ambient temperature drying [67]. During ageing, the gel structure can change due to various factors, including the dissolution of microscopic particles into bigger ones. Upon ageing, the solvents evaporate, producing gel shrinkage before a stable sol–gel network forms. Controlling the ageing duration is crucial in AG synthesis. In fact, ageing for too long causes lower pore volume, while not ageing for long enough promotes gel network instability, which can translate to gel network collapse during solvent extraction by drying methods. In this regard, the use of ionic liquids can extend the time of ageing, since their low vapour pressure inhibits solvent evaporation, while strong ionic strength accelerates aggregation. To eliminate any leftover water from the pores, the gel should be cleaned with ethanol and heptane once it has aged [68]. The final step of the sol–gel procedure for engineering AGs consists of a drying procedure, which may occur by ambient pressure drying, supercritical drying or freeze drying. Supercritical CO_2_ drying of gels is considered as the most important step of AG production, since it enables preservation of the three-dimensional pore structure, which leads to unique material properties such as high porosity, low density, and large surface area [69]. Supercritical CO_2_ drying of aged gels can be performed at high or low temperature and in autoclave at a critical level of heating or at the atmospheric pressure. It prevents the occurrence of surface tension in the gel pores and constant compression, thus avoiding the production of concave menisci and preventing pore and gel body collapse [63]. Methanol is usually used as supercritical drying solvent for AGs. High-temperature supercritical CO_2_ drying causes less shrinking of the gel. Figure 5 shows a typical flow diagram of a continuous supercritical CO_2_ drying process [69].

Estok et al. produced an AE in a kerbed mould inside a 24-ton hydraulic hot press, provided by Tetrahedron associates, completing the synthetic progression with a quick supercritical extraction process [70]. A stainless-steel mould with 16 wells with diameter 1.9 cm, depth 1.9 cm and dimensions of 11 × 11 × 1.9 cm was constructed and sealed prior to processing with gasket material. A nonstick spray was applied to the mould. Similarly, Anderson et al. used a hydraulic hot press and a bespoke steel two-part mould, which was sandwiched between Kapton and a high-temperature gasket material to achieve rapid supercritical extraction [71]. At the beginning, the mould was heated at 38 °C and stabilized by adjusting the hot press limiting strength to 94 kN for 2 to 7 h. Yoda and Ohshima modified methanol supercritical drying by adding water, nitric acid and ammonia during preparation of the SAGs from tetra-methoxy-silane, heating the autoclave to 553 °C at a rate of 40 °C per hour, and keeping the temperature constant for a further 3 h [72]. By comparing the obtained results with those observed using normal supercritical drying processes, the authors evidenced that gel-component extraction decreased, especially when water was added. The constituting particle and pore sizes increased when water or a base were added, while the use of acids increased the specific surface area (SSA) [72]. Quignard et al. prepared polysaccharide AGs by drying the obtained polysaccharidic microspheres first by immersion in a series of sequential ethanol–water baths with increasing alcohol concentrations (10%, 30%, 50%, 70%, 90%, and 100%) for 15 min each [73] and then in Polaron 3100 equipment under supercritical CO_2_ circumstances (74 bar, 31.5 °C). Although supercritical CO_2_ drying processes have several advantages, high costs, process continuity and safety strongly limit its extensive utilization. To overcome these issues, a commercially available ambient pressure drying (APD) technology was devised and proposed by Brinker et al. to produce SAGs, involving ambient pressure evaporation as the last step. APD is the most used method for preparing low-costs AGs [74]. Cai and Shan performed a two-step acid-base sol–gel synthesis of SAGs using methanol (MeOH) as solvent and oxalic acid and ammonia (NH_3_H_2_O) as catalysts, followed by ambient pressure drying [75]. Feng et al. produced SAGs with a large surface area by means of sol–gel technique using water glass made from rice husks as precursor, whose modulus strongly affected the characteristics of AGs. Also in this case, ambient pressure drying was used in the final step [76]. Also, Yun et al. prepared SAGs and dried them at ambient pressure in a furnace at 80 °C for 24 h and finally at 120 °C for 12 h [77]. Avoiding acid corrosion on the equipment during the drying process, Zhao et al. evaporated liquid in AGs at ambient pressure without the silica skeleton structure collapsing [78]. Wu et al. found that highly porous AGs, with unprecedented hydrophobic characteristics, can be achieved using fly ash and trona ore as raw ingredients and drying the wet product at ambient pressure [79]. AGs can also be dried using freeze-drying techniques, in which the liquid of the long-aged wet gel is frozen and subsequently vaporized under reduced pressure [63]. The long ageing time is necessary to achieve a full solidification of the gel network, while a solvent with a lower thermal expansion and a higher sublimation pressure should be used [63]. This approach was used by Pan et al., who employed various molar ratios (5.1–0) of MTMS/water–glass co-precursor, to create hybrid AGs of silica and carbon with low thermal conductivity and great thermal stability [80]. Experimental findings showed that the MTMS/water–glass molar ratio greatly influenced the hybrid AGs’ characteristics. After 8 h of drying at −80 °C in a laboratory freeze drier, the AGs were vacuum-dried for roughly 48 h. Additionally, glass fibre-reinforced SAGs were produced by Zhou et al. [81]. Appendix A file shows the microstructure of agarose AGs (AA-2), silica AGs (SA-4) and composite AGs (CAs), prepared by the in-situ sol–gel method, as revealed by scanning electron microscopy (SEM) images. The 3D network structure of AA-2 shows disordered and dispersed agarose nanofibers based on hydrogen bonding or electrostatic attraction between the helical agarose molecules and the flexible chains (Appendix A). Conversely, micrometre-sized SiO_2_ aggregates with the typical AG structure are observable in the pure SAs (Appendix A). Concerning CAs, during the gelation process, a gel skeleton of agarose formed first, due to its low-temperature self-coagulation, while SiO_2_ gelled in the agarose gel skeleton secondly, to provide an interpenetrating network (IPN) structure. Its SEM images show that the IPN structure was formed by a flexible agarose and a rigid SiO_2_ gel skeleton (Appendix A).

#### 3.1.1. Improved Sol–Gel Procedure by Epoxide Addition Methods

By the sol–gel approach described above, a robust assortment of metal oxide, chalcogenide and other classes of AGs has been developed. Furthermore, in the case of MOAG preparation, due to the fast reaction kinetics of this process, the key phases of the process including hydrolysis and condensation of the precursors often give dense hydroxide or oxide residues, despite ideal conditions. Consequently, the porosity of the products results significantly limited and not controllable. In this regard, it is necessary to slow down the hydrolysis and condensation rates to achieve materials with high porosity [67]. A modified sol–gel method was used by Tilloston et al. to prepare mixed lanthanide–silicate and pure lanthanide AGs from erbium, praseodymium, and neodymium chlorides, using different catalysts [82]. Particularly, to prepare mixed AGs, the authors used a two-step sol–gel method using a sub-stoichiometric amount of water in the first step, thus achieving a partially condensed silica–lanthanide precursor. Conversely, the pure lanthanide AGs were obtained directly from the chlorides using propylene oxide (PO) to neutralize the hydrochloric acid evolved during the reaction, thus slowing down hydrolysis kinetics [82]. In the example of the procedure previously described by Tilloston et al. [82], Gash et al. proposed the epoxide addition (EA) method using PO associated with the sol–gel alkoxide chemistry to prepare Fe_2_O_3_ [83] and Cr_2_O_3_ [84]. Specifically, AGs were prepared by the sol–gel procedure, using inorganic salts as precursors, and acidic hydrolysis and epoxides to induce gelation. Lower-rate kinetics were achieved, avoiding the formation of gelatinous precipitates commonly obtained by traditional sol–gel methods, thus allowing for the formation of monolithic semi-transparent AGs. Various types of epoxides were experimented on to investigate the effects of different epoxydic structures on the microstructural properties of iron oxides (Figure 6) [85].

Inspired by studies of Gash et al., many different unprecedented materials were prepared by EA methods, thus significantly escalating the range of possible monolithic AG materials. Most of the reported EA approaches involved the use of propylene oxide (PO) for gel formation by the following mechanism. In the first step (Figure 7A), the PO oxygen becomes protonated by the H^+^ of the HA catalyst necessary for the sol–gel process, while in the second one (Figure 7B), the anion A^−^ makes a basic attack on the less substituted carbon atom of PO, leading to a ring opening reaction. By this mode, acidic proton atoms are captured, thus slowly raising the pH value. Such pH change causes hydrolysis and condensation of the inorganic salts and slows down the condensation rate.

These pioneer studies of Gash et al. on Fe_2_O_3_ [83] were then progressively enlarged to Fe_3_O_4_ [86,87], β-FeOOH [85], Fe_3_C [88], Pd and K [89,90]-drugged iron oxide and mixed-metal nickel–iron oxide aerogels for oxygen evolution reaction [91]. The EA method was utilized by Wei et al. to obtain nickel cobaltite (NiCo_2_O_4_) AGs, which in fact were synthesized by a chloride-based epoxide-driven sol–gel process [92]. The as-prepared AGs were crystallized at a combustion temperature of 200 °C and were experimented on as materials to develop capacitators. They exhibited ultrahigh specific capacitances (1400 F g^−1^), excellent reversibility, and outstanding cycle stability at a relatively high mass loading of 0.4 mg cm^−2^ [92]. Al_2_O_3_-based AGs with tuneable pore size were added to the reaction mixture PO [93,94,95]. Heterogeneous AGs made of Al_2_O_3_ mixed with CuO and ZnO were proposed by Guo et al. by methanol synthesis [96], while Gill et al. co-gelled aluminum and nickel nitrate salts [97]. The EA method was also combined with non-alkoxide chemistry, thus allowing to synthesize yttria (Y)-stabilized zirconia (Zr)-based AGs [98]. The influence of various factors on surface area and morphology of such materials were investigated thoroughly by different authors [99,100], as well as alternative synthesis using different salts and alkoxides [101], including the nitric acid-assisted EA synthesis, being utilized by Zhong to prepare monolithic Zr AGs [102]. They were synthesized starting from zirconium oxychloride (ZrOCl_2_) as the precursor and PO dissolved in ethanol and mixed ethanol–water, respectively, followed by calcination at 750 °C for 2 h [102]. High surface area AGs based on SnO_2_ [103] and ZnO [104,105,106], as well as their AG composite (ZnO–SnO_2_ AG), which demonstrated efficiency in degrading rhodamine B (RhB) [107], were synthesized by different scholars. Indium-doped tin oxide AGs were obtained by Davis et al. through the EA method, utilizing a non-alkoxide procedure with glycidol, thus achieving stable networks showing a controlled pore structure and remarkable electric conductivity [107]. These AGs may meet the increasing need for porous conducting systems to develop novel batteries [108], fuel cells [109] and transparent electrodes for LCD, LED and solar cells [110,111,112]. Similarly, Correa et al. fabricated transparent, conductive AGs via EA doping tin oxide with antimony, which may be utilized as an electron scavenger in dye-sensitized solar cells [113,114]. Already before the year 2020, several materials had been used for preparing AGs via EA, which have been reported in Table 3.

The EA method offered the possibility to obtain remarkably enhanced AGs. Furthermore, the achievement of crack-free monolithic materials based on bivalent metal ions (Cu^2+^, Ni^2+^, Zn^2+^, etc.) remained an unsolved challenge. In this context, Gash et al. synthesized, for the first time, a divalent metal oxide-based AG using the EA method [135]. However, the discovery of the dispersed inorganic method (DIS) was the approach which made the obtainment of divalent metal-based AGs easy. DIS is a variation of the EA, method discussed extensively by Du et al., which consists of the addition of polyacrylic acid (PAA) and PO to the solutions of inorganic salts [136]. In the reaction mixture, PAA serves as both a dispersant via its steric hindrance and as a nucleation site for gel formation [136]. By using the DID method, bivalent metal-containing AGs with less shrinkage and stronger scaffold were achieved and three-dimensional crosslinking was easily possible [23]. As an example, Appendix A shows the surface morphologies of the Zr–Mg mixed oxide AGs prepared using the addition epoxide method, as provided by FESEM analyses. Specifically, the FESEM images refer to mixed oxide AGs with 10:0 (Appendix A), 9:1 (Appendix A), 8:2 (Appendix A), 7:3 (Appendix A) and 6:4 (Appendix A) molar ratio of Zr to Mg, respectively. Large clusters are observable for the Zr–Mg mixed oxide AGs with a 6:4 molar ratio of Zr to Mg, thus indicating that particles aggregation increased with the increase of Mg molar proportion.

#### 3.1.2. Sol–Gel Methods to Prepare Noble Metal Aerogels (NMAGs)

The sol–gel method is the most used method for preparing NMAGs under mild conditions. Two-steps and one-step sol–gel methods terminated with supercritical CO_2_ drying were reported to obtain noble metal monolithic AGs, which represent a unique class of materials having high specific surface areas and large open pores, thus being very promising in various applications such as in heterogeneous gas-phase catalysis, electrocatalysis, and sensors [137]. The only difference between the two methods is whether a separated NP colloidal solution is required [138]. The two-step method involves the reduction of metal ions to metal nanoparticles (MNPs) followed by gelation, while the second proceeds via an in situ spontaneous gelation process of metal ions [137,139]. Specifically, in the two-step strategy, citrate-capped monometallic NPs with size 3–6 nm are first synthesized via reduction of the metal precursor (HAuCl_4_, AgNO_3_, H_2_PtCl_6_, or PdCl_2_) with NaBH_4_ using trisodium citrate as a stabilizer [140,141]. Also, hollow preformed bimetallic nano-shell particles can be prepared via galvanic displacement reaction between citrate-stabilized AgNPs and metal precursors such as HAuCl_4_, K_2_PdCl_4_ and K_2_PtCl_4_, while thiolate-coated Ag nano-shell particles can be obtained via fast chemical reduction of preformed Ag_2_O NPs [142,143]. The following step involving gelation of the preformed metal NP solutions or their mixtures is induced by deliberate destabilization via typical sub-steps including solution concentration by a factor of 10–50 with polystyrene centrifuge filters or a rotary evaporator with water washings to minimize residual stabilizer and impurities. Then the gelation occurs by settling the concentrated NP solutions at room or increased temperature (323–348 K) or via addition of destabilizers such as ethanol, H_2_O_2_, etc. By such destabilization, the hydrogels formed in about 1–4 weeks, which greatly limits their applications [138]. Further, the gelation time can be shortened and gel kinetics accelerated by various novel destabilization methods developed, such as changing the synthetic parameters (e.g., temperature and disturbance) [23,141,144,145,146,147] or adding extra initiators e.g., dopamine (DA), salts, tetranitromethane (C(NO_2_)_4_), NaBH_4_, and hydrazine monohydrate (N_2_H_4_·H_2_O) [138]. A wide range of monometallic hydrogels, including Au, Ag, Pt and Pd, and multi-metallic hydrogels, including Au–Ag, Au–Pd, Pt–Ag, Pd–Ag, Pt–Pd, Au–Ag–Pt, Au–Pt–Pd, Ag–Pt–Pd and Au–Ag–Pt–Pd, which were transformed in aerogels by drying, have been prepared using this strategy, [138,140,141].

In the second strategy, hydrogels form spontaneously via in situ reduction of metal precursors with NaBH_4_ in a single step, without the preformation of adequately stabilized NPs [138]. α,β,γ-cyclodextrin (CD)-protected Pd (Pd_α,β,γ-_CD) hydrogels [148], pure Pd and Pt hydrogels, and bimetallic Pt*_n_*Pd_100–*n*_ hydrogels were prepared by this strategy [148,149]. However, due to a too-long gelation time (3–10 days) and the large amounts of organic residuals (44 wt%) in the final product, great difficulties were encountered in investigating the intrinsic activity of NMAGs. To address this issue, in situ kinetically controlled reduction and synthesis of M–Cu (M = Pd, Pt, and Au) hydrogels within 6 h via increasing the reaction temperature was pioneered by Zhu et al. [150]. Furthermore, Shi et al. employed a similar method, thus fabricating AuPt*_x_* bimetallic hydrogels for the first time at 60 °C in 2–4 h [151]. Supercritical CO_2_ drying has been demonstrated to be the most appropriate way to retain the internal structure of the hydrogel also in the dry state of AGs, minimizing the factors which may lead to the collapse of the fragile pores inside the structure, thus allowing the gel to be dried with very little shrinkage. AGs with higher surface area, an intact pore shape and a larger pore volume than those porous structures obtained using conventional drying methods are allowed by CO_2_ supercritical drying. Before supercritical drying, the water in the pores of the hydrogels is replaced by acetone and further with liquid CO_2_.

Additionally, methods such as non-sol–gel synthesis techniques of dealloying, combustion, and templating approaches, thus achieving bio-templated metal AGs composites and salt-templated structures, have been reported [139]. As an example, the TEM micrographs and mapping images of the 5% Ag/BTO-Cr010 AGs are observable in Appendix A. Particularly, Appendix A shows monodispersed BaTiO_3_ particles with a size of approximately 10 nm. Conversely, Ag nanoparticles demonstrated a particle size ranging from 10 to 70 nm. Appendix A exhibits the lattice appearance of a particle with a D-spacing of 0.284 nm and 0.236 nm, consistent with the cubic BaTiO_3_ structure (110 plane) and elemental silver (111 plane), respectively. Appendix A displays the STEM-EDX image of the AG, where the copper peaks fit the target stand, while the carbon peaks belong to the carbon adhesive. Excluding copper and carbon, Ba (14.15%), Cr (1.43%), Ti (16.17%), O (59.42%) and Ag (8.84%) were the atomic percentages of all elements. Furthermore, Appendix A shows the FESEM image of α-Ni(OH)_2_ AGs prepared using a two-step sol–gel method, followed by a freeze-drying technique (Appendix A) and of the NiO/Ni AGs. The latter AGs, which demonstrated good porosity, were obtained by the synthesized α-Ni(OH)_2_ AGs, upon annelation at 400 °C. The network-like continuous structures of such calcinated AGs looked to be made of nanoflakes of Ni (Appendix A). SEM micrographs of nickel AGs [α-Ni(OH)_2_] and annealed samples (NiO/Ni) showed flaky nanoporous, wafer-like nanoflower structures. The α-Ni(OH)_2_ samples demonstrated considerably high aggregation, while upon annelation, the resultant NiO/Ni samples showed more open structures with less aggregation.

#### 3.1.3. Non-Sol–Gel Methods to Prepare Metallic Aerogels

Dealloying and combustion synthesis is a non-sol–gel method to achieve metal nanofoams and aerogels, addressing the challenges of synthesis time and shape control of the one- and two-step aerogel synthesis reported above. Additionally, a bio-templating strategy has allowed strict control over nanostructure length and diameter, as well as overall monolith shape. Also, salt templating is an emerging synthesis method for high-surface area, porous, and scalable metal-based AGs [139].

##### Dealloying and Combustion

Dealloying is a general term for a corrosive process that selectively removes one or more components from an alloy, leaving the remaining components to evolve spontaneously into a porous structure and was the first method employed to synthesize porous noble metal foams [139]. Subsequently, this method has been employed to prepare films, foams and porous nanoparticles [152,153,154,155,156,157]. Alloys based on the association of Ag and Au were exploited for selective dealloying, since silver is easily dissolved, while Au atoms tend to aggregate, thus forming a relatively homogeneous 3D porous structure. Among the disadvantages of this technique, the small number of alloy combinations available and the impossibility to scale well beyond 1 mm are comprised [158], while sizes range down to approximately 10 nm resulted in specific surface areas lower than 10 m^2^/g. Collectively, despite the fact that this technique may be promising when in combination with other synthetic strategies, limitations for scale-up and practical device implantation reduce its attractiveness. Combustion synthesis has also been used to prepare gold and palladium aerogels through the ignition of metal complexes with highly energetic ligands [158,159]. Specific surface areas for Au and Pd monoliths prepared via combustion synthesis are reported to be 10.9 m^2^/g and 36.5 m^2^/g, respectively, suggesting a similar challenge as dealloying in attaining small-scale feature sizes. As an example, Appendix A schematizes a process used to prepare cobalt porous gold nanoparticles, including a dealloying phase (A) and the secondary electrons SE-SEM image of the porous gold nanoparticles (NPG), obtained by dealloying the amorphous precursor. In particular, the morphology of the freshly prepared dealloyed NPG appeared characterized by islands of pure Au with sub-micrometric size and irregular shape, while a well-defined porous structure develops among the micro-islands. Conversely, Appendix A exhibits the SEM imageries of biogenetic MnO/C (CMB), namely B800 (Appendix A) and MnO/C/NiO (CMB-Ni) (Appendix A) porous composites. They were obtained by combusting the MnO_2_/bacteria (BMB) and the MnO_2_/bacteria/Ni (BMB-Ni) porous composites, prepared by a bio-templated method based on *Pseudomonas putida* cell-surface display technology. Combustion was carried out in a tube furnace at 800 °C under an Ar atmosphere. The organic carbon of the organisms were decomposed during the combustion process, thus generating the reducing atmosphere, which reduced MnO_2_ to MnO. Meanwhile, the residual inorganic carbon was converted into a carbon coating and electroconductive carbon matrix. After decomposition, cavities and macropores were formed in place of organic matter.

##### Bio-Templating

Bio-templating is an approach which allows us to achieve materials with biological morphology and having different compositions and shapes, utilizing material chemistry. Specifically, bio-templating exploits biological molecular structures, such as microorganisms, viruses, and biomolecules such as DNA, cellulose, and proteins as shape templates to synthesize nanostructures via binding preformed nanoparticles, vapour deposition, or electroless depositing of metals onto the template surface [160,161]. Using this technique, several nanomaterials and porous films have been prepared, but its use to achieve noble metal aerogels is still in its infancy. As an example, Appendix A shows the SEM images of biogenic MnO_2_/bacteria (BMB) porous composites prepared by a bio-templated method based on *Pseudomonas putida* MB285 cell-surface display technology. The aggregates appeared as unbalanced spherical secondary particles made of exfoliated Mn oxides stuck to bacteria surfaces and dispersed spherical Mn oxides. The aggregate structure was dense and preserved its intact morphology even at an SEM operated voltage of 200 kv. The aggregates showed a relative uniform size of 20–30 µm.

##### Salt Templating

Dealloying and gravity-driven gelation approaches can provide noble metal foams and NMAGs as thin films with characteristics which make them suitable as electrodes [153,162]. Easy scaling-up, short time, and low costs are the main advantages to prepare thin films, due to the shorter diffusion path to remove synthesis by-products and conduct solvent exchanges. Burpo et al. used insoluble salt needles as templates for porous macro-tubes and macro-beams that may be assembled into arbitrary shapes, such as thin films, and with tuneable densities [163,164,165,166]. The reduction of salts formed with varied concentrations of [PtCl_4_]^2−^ and [Pt (NH_3_)_4_]^2+^ ions, namely Magnus’ salts [139], used as insoluble precursor salt needles, yielded hierarchically highly porous macro-tubes with square cross sections and contour lengths consistent with the salt templates (SEM). SEM images showed porous sidewalls comprising Pt nanoparticles (NPs) (~100 nm in diameter), in turn comprising textured nanofibrils with an average diameter of 4.9 ± 0.7 nm. Reduction was carried out with 0.1 M NaBH_4_ at 1:50 (*v*/*v*) salt needle. It has been reported that two [Pt (NH_3_)_4_]^2+^ are necessary in solution to maintain a charge balance for each [PtCl_4_]^2−^ ion reduced. X-ray photoelectron spectroscopy and X-ray diffractometry analyses indicated the presence of platinum structures with no oxide content, while electrochemical impedance spectroscopy (EIS) demonstrated a specific capacitance of 18.5 F/g and an electrochemically active surface area (ECSA) of 61.7 m^2^/g. Cyclic voltammetry (CV) showed characteristic hydrogen adsorption and desorption and platinum oxidation–reduction peaks [166]. Combining different square planar noble metal salts to form Magnus’ salt derivative needles as salt template allowed the achievement of bimetallic and alloy noble metal-based materials. In this regard, Burpo et al. studied the salt-template reduction–dissolution mechanism for the formation of platinum–palladium hierarchical metal nanostructures [164]. Particularly, the addition of [Pt(NH_3_)_4_]^2+^ ion solutions with varied concentrations of [PtCl_4_]^2−^ and [PdCl_4_]^2−^ anions provided bimetallic salt templates with lengths between 15 and 300 μm, which, upon chemical reduction with NaBH_4_, formed porous Pt-Pd macro-beams with square cross sections that were in compliance with the length of the initial salt template. Porous sidewalls and internal structures comprising primary Pt and Pd NPs of 8–16 nm in diameter, or fibrils 4–7 nm in diameter, depending on the ratio of platinum to palladium ions in the salt template, were observed for macro-beams, each beam representing an AG structure. ECSA values were in the range 23.2–26.7 m^2^/g. Despite these values being almost half the BET-determined specific surface area values of Pt-Pd bimetallic aerogels synthesized by Bigall et al. using the two-step gelation approach [140], in the salt template method, the formation of the salt template is instantaneous, and their electrochemical reduction is very rapid. As an example, Appendix A shows the SEM images of biomass-derived porous carbon materials, with a good balance between high specific surface area and mesopore volume, prepared via a molten chloride salt templating technique and successive KOH activation (MHPC-700, MHPC-800 and MHPC-900) (Appendix A). For comparison, Appendix A shows the SEM image of the carbon sample NHPC-700 pre-carbonized in nitrogen atmosphere without molten salt. A sheet morphology is prominent in MHPC-700, while a thick carbon block dominates in NHPC-700, demonstrating that the sheet structure formed during the pre-carbonization process in molten salt. Due to the role of molten salt generating mesopores, in the pre-carbonized product, which can further serve as the active sites for the KOH activation to form micropores in the final carbon material, the mesopore–micropore structure of the porous carbon can be tuned by changing the pre-carbonization temperature. 

### 3.2. Nanoparticle-Based Routes to Aerogels (AGs) by Wet Sol–Gel Processes

Over years of studies in nanotechnology, a robust arsenal of NPs owing different sizes and shapes, which can be used as versatile building blocks with distinct properties and functionalities to achieve AGs, is now available. Various types of NPs can be assembled as LEGO bricks in a bottom-up process [167] to form AGs monoliths conserving the NP properties in a nanostructured macroscopic bulk material. Despite the as-prepared AGs not possessing high porosity, high surface area and low density, they encompass new nonpareil functionalities, such as super-paramagnets [168], ferroelectricity [169], luminescence [170], (photo) catalytic activity [171,172] or electrical conductivity [20,173], which were properties of their NP precursors, thus being very interesting as batteries [174], fuel [175] or solar cells [113]. Following this, a general overview of the different steps of assembling preformed NPs into AGs has been made available. The gelation of preformed nanocrystal building blocks dispersed in high concentration in solvents to form three-dimensional macroscopic AG monoliths usually occurs by controlled destabilization. In most systems, this phenomenon leads to the assembly of the NPs in gels at random crystallographic orientation (as is the case of metal chalcogenides AGs) or by an oriented attachment mechanism (as is the case of TiO_2_ and SnO_2_ AGs). Finally, the dispersions of anisotropic NP building blocks can often be destabilized through mild centrifugation (as is the case with WO*_x_* and Y_2_O_3_ AGs). Collectively, the entire procedure thus requires the synthesis of the nanosized building blocks by wet chemical processes to have good control over particle size, size distribution and shape, as well as over the surface chemistry, thus preventing or minimizing agglomeration for good re-dispersibility in the next step. Available wet synthetic methods encompass aqueous [176] and nonaqueous sol–gel processes [175,177], polyol route [178], hot-injection [179], heating-up method [180], hydro- and solvothermal processing [181], etc. Once synthesized, the building blocks need to be immersed in solvents to provide dispersions concentrated enough to reach a percolation threshold during gelling, but not excessively high to avoid aggregation. For stabilizing the dispersion, scientists use brushes, surfactants or electronic charges to avoid spontaneous aggregation. To this day, a useable protocol to prepare perfectly concentrated colloidal NP dispersions is still non-existent despite extensive research. All specific different systems have an exclusive recipe of stabilizing strategies and solvents, as well as a unique equilibrium of attractive and repulsive forces between particles, which determine if the particles will give a stable dispersion or will undergo coagulation [182,183,184]. A pivotal step towards AG formation is the controlled and efficient destabilization of the dispersions. Among the most adopted destabilization approaches, photochemical treatment [185], temperature change [173], sonication [169], chemicals adding [170] or the use of solvents [173] to remove stabilizing ligands from the surface of the NPs [172] or neutralize surface charges by changing the ionic strength [142] or pH [186] of the media are preferred. The incorrect balance between attractive and repulsive forces drive the prevalence of attractive interactions [187]. This event leads NPs to lose their hard sphere-like comportment, thus becoming more “sticky” and colliding/fusing together [188]. To avoid flocculation and sedimentation in place of AG formation, it is of paramount importance to control the rate of the destabilization process [187,189,190]. In most systems, NPs fuse upon contact at random orientations and atom diffusion can occur, while TiO_2_ and SnO_2_ systems undergo oriented attachment upon controlled conditions. The last step consists of the supercritical drying of the wet gel to obtain the NP-based AGs. Building blocks of a few nanometres can assemble to provide centimetre-sized macroscopic structures, thus seeming to be a particularly impressive process considering that it ties seven orders of magnitude in length scale. As an example, Appendix A shows the SEM image of hybrid AGs combining collagen (C) and chitosan (CH), prepared without nanoparticles (Appendix A, Ref-AG) and using chemical (Appendix A, Ch-AG) and green (Appendix A, Gr-AG) iron oxide nanoparticles dispersions, previously synthesized, as building blocks. Nanoparticle dispersion in acetic acid in a test tube was placed in an ultrasound machine for 15 min to facilitate their dispersion. Then, they were added to biopolymer solution after their centrifugation and mixed before the freezing step. The three AGs showed an irregular structure, characterized by the presence of micro- and macro-pores. The Ref and Ch AGs demonstrated very similar laminar structures (Appendix A), with a defined directionality. The Gr AGs also exhibited a laminar structure, but not directional, and displayed greater heterogeneity (Appendix A). Regarding porosity, the Ref and Ch AGs displayed a multitude of pores, exhibiting heterogeneous sizes and shapes (with a mean porosity of 46.5 ± 15.8 nm and 29.4 ± 17.4 nm, respectively) and a relatively uniform dispersion. Conversely, Gr AGs were devoid of pores, with the existing ones being uniformly minute.

The following Figure 8 summarizes the main synthetic methods to prepare AGs.

## 4. More in Deep into the Most Patented Classes of Inorganic Aerogels

In this section we have deeply studied the two most experimented and patented classes of inorganic AGs, such as silica and metal oxide AGs, in terms of main properties and applications.

### 4.1. Silica-Based Aerogels (SAGs)

Silica-based aerogels (SAGs) present a distinct nanoporous network made of interconnected silica nanoparticles (SiNPs) and a high-volume of nanosized pores, which have attracted the interest of many researchers. Silica was the first material produced in the form of AG, and it became the most extensively studied system in the community, and nowadays, most publications on inorganic AGs are dedicated to silica-based AGs. The most used precursor material to prepare SAGs is silica, followed in order by Si(OCH_2_CH_3_)_4_ (tetra-ethoxy-silane), CH_3_Si(OCH_3_)_3_ (methyl tri-methoxy-silane), sodium silicate, Si(OCH_3_)_4_ (tetra-methoxy-silane), Cl-Si(CH_3_)_3_ (chlor-trimethyl silane), SiO_n_(OC_2_H_5_)_4_ (poly-ethoxy di-siloxane) and silicon [2]. AGs from tetra-methoxy-silane and poly-ethoxy di-siloxane have much lower thermal conductivity than those from tetra-ethoxy-silane precursors [191]. Tetra-ethoxy-silane precursor is the best option to achieve high-quality clear aerogels. They are achieved via the “sol–gel” process, which, as above reported, is a popular and effective method for making the various types of aerogels [192]. Chemistry and kinetics of sol–gel reactions to prepare SAGs are well known and controllable and silica offers extensive potential for surface functionalization [171]. The steps in the sol–gel process include precursor preparation and formation of colloidal solutions (sol) by their acidic or basic hydrolysis followed by gelation through stable covalent crosslinking or weaker intermolecular interactions, which can undergo reversible sol–gel and gel-sol transitions under external conditions [193] and ageing. During ageing, the just-formed gel network is allowed to expand with the help of managing factors like pH level, temperature, and time. Ageing parameters, such as Ostwald ripening, coarsening, sintering, and syneresis, have a significant impact on the physical features of the product, particularly in terms of fortifying the fragile structure of the gel which is formed by improving siloxane bonds between particles before drying [194]. Finally, a drying procedure like supercritical CO_2_, freeze drying, or ambient pressure drying and densification conclude the process [192]. SAG hydrophobicity or hydrophilicity mainly depend on the synthesis process or on the surface silanol groups [195]. SAGs on the market find applications in high-tech engineering such as thermal insulation, separation, coatings, sensing, and catalysis due to their unique features [195]. Additionally, they find application in biomedical sectors for drug delivery, tissue engineering and regenerative medicine [196,197,198]. Recently SAGs have been applied for environment remediation, air purification, CO_2_ capture and VOC removal, as well as in water treatment, to remove oil and toxic organic compounds and heavy metal ions [195]. Table 4 collects fundamental physical properties of SAGs, while some of the important SAGs along with their mechanical properties in terms of Young’s modulus, bulk density and/or compressive stress are listed in Table 5.

For further information, refer to other publications and reviews covering the recent advances in SAG research [13,209,210,211].

#### 4.1.1. Main Properties of SAGs

SAGs are endowed with unprecedented properties which give them broad utility in various high-performance applications. SAGs have a typical microstructure characterized by particles with a size of 2–5 nm and ultra-fine pores, sized in the range of 50–100 nm, depending on the specific synthesis conditions, tailored during their production. Unfortunately, despite SAGs being admired for their exceptional thermal insulating properties, mainly depending on their huge porosity, they frequently display poor mechanical strength. Their density affects their brittleness and ductility, with higher densities being characteristically associated with augmented fragility, which powerfully hampers their extensive industrial utilization [212,213]. This weakness essentially depends on their gentle “pearl-necklace” microstructure, particularly susceptible to the interparticle collar regions, which impairs structural integrity when mechanically stressed [214]. Polyethylene glycol (PEG) is typically integrated into the aerogel matrix for enhancing mechanical properties of SAGs. Specifically, by modifying the pore sizes, their overall durability can be enhanced, making them less prone to mechanical failure under stress [215,216]. Also, papers have reported that a lower pH and extended ageing can lead to smaller pores and a denser particle network, thus enhancing the mechanical properties and thermal resistance of AGs [217,218]. Moreover, it has been demonstrated that the introduction of carbon-based materials into SAG networks improves SAGs’ mechanical strength and elasticity, thus leading to advanced resistance to compressive and tensile stresses, without prejudicing the SAGs’ lightweight characteristics [219]. Also, by combining SAGs with proper polymers and ceramics, which act by supporting their structure, the toughness and durability of SAGs can be improved. Using this approach, materials more suitable for uses in construction and industrial settings, where capacity in tolerating fluctuating environmental conditions and mechanical loads are required, can be achieved [220]. Supercritical CO_2_ drying methods that maintain AG microstructure by avoiding pores collapse can greatly contribute to boosting the mechanical profiles of DAGs [221]. The hybridization of SAGs with more robust materials through co-gelation processes including organic materials, such as fibres and polymers, has led to reinforced monolithic structures, with improved mechanical properties suitable for large-scale industrial applications, as extensively discussed in the notable review by Xue et al. [222]. Patil showed experimentally that the integration of just 2% double-walled carbon nanotubes (DWCNTs) into DWCNTs meaningfully augmented their mechanical properties, tripled their elastic modulus and markedly enhanced tensile strength [223]. The reinforcement of SAGs with polyacrylonitrile fibres enhanced their durability, reusability and Young’s modulus, thus making them exceptionally effective in oil adsorption. Conversely, the introduction of polyaniline nanofibers increased SAGs flexural strength while maintaining their essential electrical properties, thus providing materials with more potential in electrical applications, including the development of chemo-resistor sensors [224]. Polymer-reinforced SAGs that exhibited increased elastic recovery were synthesized. Pettignano et al. explored alginate as a reinforcement material for SAGs, finding enhanced mechanical properties [225]. Cao et al. experimented with the incorporation of polyurethane foams in SAGs, obtaining composites that retained high porosity, while demonstrating ameliorated deformability and mechanical resistance. Such materials have a wider range of applications, spreading from construction to aerospace sectors, where both lightness and durability are essential [226]. Advanced manufacturing techniques like direct ink writing allowed the obtainment of objects with customized mechanical properties and low thermal conductivity for high-performance thermal insulation applications [227]. SAGs also possess significant optical, acoustic and low thermal conductivity properties. SAGs inherent transparency, which can be effectively controlled through adjusting synthetic conditions, including pH values, ageing time, and the sorts of precursors [228], make them particularly appealing for applications in optical fibres and solar collectors, where precise light manipulation is necessary. Additionally, they attract interest for use in improving the efficiency and performance of photonic devices, which are of paramount importance in telecommunications and information processing technologies [229,230,231]. The low refractive index of SAGs is perfect for optical fibres, which need minimal light loss, to maintain the integrity of signal over extended distances. In this context, SAGs appear as excellent candidates for inclusion in optical devices, including lenses and light guides [232,233]. Also, SAGs are endowed with controlled light scattering, thus being appropriate for architectural applications and for modern, energy-efficient building designs. They are auspicious for developing translucent insulating windows, which permit natural light to diffuse, while energy efficiency and thermal comfort can be preserved [234,235]. Additionally, as SAGs are gifted with both low density and high porosity, this facilitates their use in high-precision imaging and sensing devices for both terrestrial and aerospace requests [236,237]. Optical components have been industrialized for outdoor applications, which can tolerate environmental fluctuating conditions, such as temperature variations and mechanical stress, without prejudicing their optical integrity [238]. On the other hand, SAG acoustic properties mainly derive from their lightweight, porous construction and larger particle sizes, which successfully absorbs sound by decreasing sound velocity. In this regard, SAGs could be supreme for soundproofing applications in various settings, including recording studios, automotive manufacturing, industrial settings, vehicles cabins, engines and aerodynamic components, etc., where the control of environmental clamour is crucial [239,240,241,242,243,244]. The application of SAG-based acoustic insulators can significantly improve workers quality of life and align with environmental sustainability objectives by helping in the engineering of more noiseless and energy-efficient vehicles [245]. Also, SAGs are excellent materials for the construction of thermal-insulating divider walls or ceiling tiles, while reducing buildings structural load [246,247]. As in the previous case, hybrid materials superior in sound absorption while maintaining structural integrity have been developed by the association of SAGs with polymers and fibres [248]. Particularly, when SAGs were combined with shape-memory polymers and phase-change materials, smart acoustic systems were achieved which were capable of adapting their noise insulation properties in response to environmental changes, such as temperature and pressure fluctuations. Such products are particularly impactful in dynamic industrial environments where ambient conditions vary significantly [249]. Moreover, disrupting sound wave propagation using SAGs could be particularly advantageous in environments where precision in sound control is critical, such as in acoustic laboratories or in high-fidelity audio production sceneries [250]. Due to their low density and high porosity, SAGs may allow for a better transmission and reception of ultrasonic waves, thus being appealing for ultrasonic applications, where they can revolutionize medical imaging techniques, such as ultrasonography [251]. Due to the scarce conduction paths in their typically air-filled pores and the complex structural system that hampers heat transfer through both convection and radiation, SAGs possess a low thermal conductivity, typically around 15 mW/mK. For this reason, SAGs are valuable in sectors that request high-performance insulation solutions, including the construction and aerospace industries [252,253]. SAGs are applied in the manufacturing of insulating clothing for extreme environments and in the construction of energy-efficient windows that reduce heat loss without losing transparency. SAGs can assure thermal protection for spacecraft and satellites, thus shielding instruments and crew from the strong thermal variations characterizing the outer space [254,255]. The incorporation into SAG scaffolds of graphene and carbon fibres, which are materials with complementary thermal and mechanical properties, led to composite AGs with maintained low thermal conductivity and improved strength and durability [256]. Such carbon-improved SAGs may be applied in the building industry for the development of AGs-imbued plasters, as well as in the field of personal protective equipment [257]. The capability of SAGs to shield against heat while being lightweight and thin can allow us to realize compact electronic assemblies, where the management of heat is pivotal for the maintaining of device performance and its longevity [258]. Collectively, SAGs are versatile materials, which can play a critical role in the advancement of various technological and industrial sectors. SAG attributes are not only crucial for their current uses, but also for the revolution and development of new future applications based on their unique characteristics. Wu et al. [259] and studies from other authors [260,261] have evidenced that pore size, density and insulating properties of SAGs can be tuneable in regulating the content of tetramethyl-orto-silicates (TMOS) used as precursor, thus achieving SAGs custom-made for specific uses, ranging from energy-efficient building materials to components in electronic devices. Similarly, Mandal et al. demonstrated that altering the TMOS content significantly influenced the light transmittance of the SAGs [262]. In fact, by tuning TMOS concentration, the authors optimized the optical transparency, achieving AGs suitable for applications in specialized environments, requiring high-level light transmission [263,264]. Bi et al. have reported that SAGs can reach sound velocities in the range 100–300 m/s, which represents a notable reduction in the sound velocity in air, which is approximately 343 m/s [265]. Malfait et al. [266] as well as Gu and Ling have demonstrated that, despite slightly less insulating, because of a minor porosity, denser AGs can provide improved structural integrity [267]. This compromise between thermal insulation and mechanical strength is essential for designing AGs with specific applications, where both characteristics are important [268]. Additionally, progresses in bio-templating techniques to prepare SAGs have allowed us to synthesize AGs with biomimetic scaffolds which imitate natural structural systems, such as the fibrous conformations of plant stems and/or the complex lattice structures of bones. The obtained materials can boast enhanced mechanical resilience and flexibility, thus being exploitable in more dynamic environments where traditional AGs would fail [269].

##### Case Studies Concerning SAGs

Based on the above-discussed properties, SAGs can have various applications, including science-based and engineering-based uses. In this section the main applications of SAGs have been reported and discussed following the scheme reported in Table 6. Table 6 evidences the direct relationship between the SAGs physical characteristics and their application in arenas that profit from their unique properties such as thermal and acoustic insulation, as well as optical transparency, as reported in the previous section [5,270].

##### SAGs for Thermal Insulation, Glazing Systems and Solar Collectors

As reported in Table 6 (first row), SAGs, known for their unprecedented thermal insulation features, are increasingly being integrated into numerous engineering sectors, including the building construction and aerospace sectors, where they have been utilized for thermal shielding in spacecraft. SAGs outperform traditional insulation materials due to their higher R-value per inch. NASA used SAGs to insulate and protect the delicate instrumentation of the Mars Rover during its mission on the Martian surface [271]. Another notable example of using SAGs to insulate buildings for energy saving is the Alaska Building in Manchester, UK, in which SAG plaster application resulted in a reduction in energy consumption by over 35% [272]. According to several authors, SAGs could revolutionize the glazing industry, thus significantly reducing energy consumption, while maintaining light transmittance. In the relevant review on SAGs by Firoozi et al., it has been reported that filling double-glazing systems with SAGs lowers the use of energy to heat and cool by over 50%, with respect to traditional glazing, while maintaining indoor comfort [199], thus underscoring the potential of SAGs in green building practices. Due to their exceptionally low thermal conductivity, both monolithic and granular AG-based glazing systems are documented for their higher performance over conventional ones. For instance, Li et al. developed a glazing system encompassing two 6 mm glass panels with AG powder, sandwiched in between and sealed with aluminum brackets [273]. Authors demonstrated that, if compared to traditional single-glazed systems, this AG glazing ensured a more relaxing indoor visual environment, while achieving notable energy savings via reducing air conditioning requirements and enhancing indoor thermal comfort. Liu et al. manufactured a glazing system comprising two 16 mm sheets of transparent plastic with a granular SAG filling, which demonstrated a solar energy transmittance of 35% and a heat transfer coefficient of 0.5 W/mK [274]. Later, the same authors studied the relationships between the size of SAGs granules and the efficiency of SAG glazing units in reducing heat loss and decreasing heat transmittance, finding that larger granules could reduce them by 58% and 38%, respectively [275]. Belloni et al. developed a SAG solar collector with a solar transmittance of 17–35% and a low thermal conductivity coefficient of 0.4 W/m^2^K, thus showcasing substantial improvements in energy efficiency [276]. To study the energy performances of glazing systems filled with different SAGs, Liu et al. calculated their solar extinction coefficient by Mie scattering and Monte Carlo method, discussed the influences of porosity and nanoparticle’s size, calculated their solar heat gain coefficients versus incidence angle and simulated their energy performances using a dynamic heat transfer model [275]. The results indicated that porosity of monolithic SAGs influences the solar extinction coefficient more than diameter and that the reciprocal effect between the porosity and the diameter is negligible [275]. Additionally, it was assumed that the SAGs having small nanoparticles and low porosity will lead to a better energy conservation performance in cooling dominated region. Also, it has been reported that creating an airtight seal by evacuating the air from double-glazing units by using vacuum insulation glazing (VIG) technology improve their thermal performance and insulation effect while reducing thermal loss, representing nearly 40% of a building’s heat dissipation [277]. In this context, due to the superior thermal performance of SAGs, many researchers have embedded them into double-glazing windows, thus developing hollow silica-based double-glazing windows [278,279,280], which were subsequently subjected to air vacuum process. Schultz et al. applied a VIG to monolithic SAGs between two glass panes, whose vacuum was 1000–10,000 times less intense than in standard VIG, achieving glazing systems with superior insulation, regardless of the low vacuum [281]. Neugebaurer et al. conducted a study and developed a technique to compact the bed of granular SAGs having thermal conductivity of 24 mW/(m K) (when uncompacted) to reduce it by 13 mW/(m K) (after compaction) [282]. The authors found that there is an optimum level of compaction to minimize the thermal conductivity, above which the contact area between the granules increases and the granules densify, increasing conduction through the solid [283]. Despite these unequivocal advantages, challenges such as high-cost production and mechanical fragility strongly hinder their broader application in civil engineering fields.

Anyway, Karim et al. successfully developed SAG-incorporated mortars (namely AIMs in the study) achieving amphibious materials, with improved binding properties, thus effectively addressing these limitations. Like Karim, Jia et al. developed AIMs containing from 50% to 70% SAGs by volume, with decreased compressive strength (from 20 MPa to 4 MPa) [284]. Appendix A shows the appearance of phenolic resin AGs (PR) and boron-modified phenolic resin AGs containing 5 (BR_5%_), 10 (BR_10%_) and 20% (BR_20%_) boron, before and after high-temperature heat treatment. While PR samples showed remarkable volume reduction after heat treatment, and important cracks on the surface, thus indicating that PR experienced severe gas escape and volume contraction during heat treatment, the BR_5%_ and BR_10%_ samples showed relatively good performance after heat treatment, with intact dense surfaces and fewer cracks. The overall structural stability of these samples was high, and no serious cracks or damage occurred. The thermal stability of these two samples in air was significantly improved, indicating that moderate boron content is helpful in improving the thermal stability of AGs, thus providing excellent materials for thermal insulation. In contrast, the BR_20%_ sample showed significant surface collapse after heat treatment, with larger areas of depression and more cracks than the BR_5%_ and BR_10%_ samples.

##### SAGs for Energy-Efficient Building Solutions and Acoustic Insulation

In addition to the glazing industry, SAGs have been used to develop AG-based plasters, renders, roof panels, insulation blankets and vacuum-insulating panels, mainly motivated by the global demand for energy-efficient building solutions to cut costs and mitigate environmental impact [285,286,287,288]. In this context, the United States spend 19% of their total energy use to heating and cooling buildings, while regions like Romania lack modern insulation, thus urgently demanding more efficient insulating materials, possessing mechanical strength, longevity, cost-effectiveness and environmental sustainability [289,290]. Song et al. successfully incorporated 10% SAGs by volume into geopolymer foam AGs renders, achieving ultralight materials with superior thermal and acoustic insulation capacities [291]. Fantucci et al. have developed plaster coatings encompassing 90% of SAG pellets, which demonstrated a thermal conductivity 10 times lower than that of conventional plasters. (0.05 W/mK vs. 0.5 W/mK) [292]. Similarly, Zhu et al. used SAG cement mortar for coating concrete tunnels, successfully augmenting protection against fire and preventing humidity-induced explosive spalling [293]. Conversely, Wang et al. developed phosphorus-doped molybdenum disulfide/epoxy SAGs, demonstrating enhanced fire resistance, thus being notable materials in critical applications such as battery enclosures, where preventing thermal runaway is a critical safety concern in the energy sector [294]. Ganobjak et al. successfully replaced traditional silica sands with SAG masses to generate lightweight cement renders for thermal insulation with significantly lower thermal conductivities of 0.085 W/mK and densities of 0.41 g/cm^3^ compared to traditional cement renders [295]. Also, Rostami et al. studied the incorporation of SAGs particles into concrete media, finding that they can partially or fully replace orthodox aggregates, while improving the thermal insulation effectiveness of old-style concrete products [296]. With the aim of externally waterproofing, while allowing internal moisture to escape, Zhang et al. have recently described a practical application of SAG render, used in the form of a 6 mm thick filler in a building constructed in 1989 [297]. This application effectively reduced the wall’s thermal conductivity from 1.0 W/mK to 0.3 W/mK, thus establishing the practical benefits of AGs in real-world applications.

##### SAGs for Environmental Remediation

Row two in Table 6 shows that SAGs find application also for environment remediation, because of their superior absorption capabilities, mainly due to their hydrophobic nature of their inner scaffold and high porosity [298,299,300]. On the other hand, SAGs possess hydrophilic surfaces, due to the presence of hydroxyl groups, which, despite facilitating the attraction and retainment of water molecules from the air, facilitating adsorption, pose risks of structural collapse due to capillary stresses. Çok et al. and Renjith et al. developed methods to chemically modify SAGs, making them more hydrophobic and effective for adsorbing non-polar substances such as oils and organic solvents [279,301]. These modified SAGs have shown strong efficiency in the adsorption of oil spills in marine environments, thus limiting the devastating impact that oil spills can have on aquatic ecosystems. As absorbent materials, SAGs are highly effective in absorbing pollutants without saturating quickly and, particularly, in oil spill cleanups. Notably, SAGs were used to absorb oil efficiently, during the cleanup efforts in the Gulf of Mexico, significantly mitigating environmental damage [302]. Saharan et al. synthesized SAGs starting from MTMS as precursor, under acidic conditions, achieving absorbent materials with superior oil adsorption capacities, underscoring the influence of synthesis conditions on AGs efficiency [303]. SAGs have showed efficiency in carbon dioxide, argon, nitrogen and oxygen adsorption, from the air, mainly if modified with amine groups. Amine-modified SAGs have been shown to adsorb considerable amounts of CO_2_, thus being valuable in air purification and greenhouse gas reduction [304]. Appendix A shows the schematic illustration of the preparation process (Appendix A) and use for polluted air filtration (Appendix A) of biomimetic grooved ribbon AGs inspired by the structure of *Pinus sylvestris* var. mongolica needles (UPG AGs). The AG, namely UPG-10, demonstrated the best performance, achieving a filtration efficiency of 99.24%, with a pressure drop of 95 Pa. Notably, it exhibited a remarkable dust-holding capacity of 147 g/m^2^, and its NH_3_ adsorption capacity reached 99.89 cm^3^/g, surpassing PG AGs by 31.46 cm^3^/g. Additionally, UPG-10 exhibited outstanding elasticity, maintaining over 80% of its original shape after 30 compression cycles. This biomimetic AG represents a promising solution for air purification, contributing to improved agricultural efficiency and environmental sustainability.

##### SAGs for Cherenkov Counter and High-Energy Physic Experiments

According to the third row in Table 6, due to their unique optical properties, SAGs can find application in Cherenkov counters to detect high-speed particles and for the precise measurement of the Cherenkov effect, essential in high-energy physics experiments. Yan et al. used SAG samples of different densities and thicknesses to develop a Cherenkov radiator for the measurement of the time spectrum of high-energy pulsed electron sources [305]. In his relevant review, Kharzheev discussed Cherenkov counters based on SAGs, reporting cases showing that they notably enhance particle identification in accelerator experiments [306]. Cherenkov counters that employ SAGs as radiators and photodetectors are often used to identify subatomic charged particles, including electrons, protons, and ions, to measure particle velocities in accelerator-based particle- and nuclear-physics experiments and in space- and balloon-borne experiments in the field of cosmic-ray physics [307]. In the past, Tabata et al. developed hydrophobic SAGs tiles with large-area (18 cm × 18 cm × 2 cm), uniform density and high refractive index (n ∼ 1.05) which were installed in a Cherenkov detector for the next-generation accelerator-based particle physics experiment Belle II [308]. By optimizing the supercritical CO_2_ drying, the initially observed cracking has been eliminated from the prototype materials [308]. More recently, the same authors engineered a SAG ring-imaging Cherenkov detector to be used in the spectrometer of a planned balloon-borne cosmic-ray observation programme, HELIX (High Energy Light Isotope Experiment) [309]. In the last-step production, a pin-drying technique, which led to fabricate 40 out of the 48 tiles with no tile cracking [309] was used, while a waterjet cutting device was employed to fit the aerogel tiles into a radiator support structure [309]. In this field, particle identification power of SAG RICH detectors strongly depends on optical properties of radiators. Barnyakov et al. demonstrated that among the several procedures of polishing of SAGs tested to improve their optical properties, the best performing was that used natural silk tissue, allowing the production of radiators for the Focusing Aerogel RICH detectors [310].

##### SAGs for Biomedical Applications

As reported by García-González et al., AGs are substantially contributing to nanomedicine, where they are mainly experimented as drug delivery systems, due to their exceedingly porous structure, which allows fast drug loading and controlled release [311]. Esquivel-Castro et al. reported that the intimate architecture of SAGs can encourage both rapid drug loading and targeted release, thus meeting critical challenges in the delivery of therapeutics, such as low drug absorption and early degradation by gastrointestinal (GIT) enzymes [312]. Moreover, chitosan (CH)-based SAGs have been experimented in tissue engineering and regenerative medicine, where they facilitated enhanced drug delivery mechanisms, bioimaging and biosensing, due to their biocompatibility and tunability [21,63,313,314]. Additionally, the utilization of SAGs for wound healing application has been extensively reported, due to their high porosity and specific surface area that allows gas exchange and rapid absorption of a large amount of exudate as well as loading bioactive molecules [315,316]. Chen et al. developed a multifunctional SAGs (CG@DA@VEGF) by a simple and eco-friendly freeze-drying process combined with harmless EDC/NHS, which were used as crosslinking agents, while CH and dopamine-grafted gelatine were the raw materials [316]. In vitro experiments showed the notable antibacterial and antioxidant effects of CG@DA@EGF, as well as supreme cytocompatibility. In vivo experiments demonstrated the capability of CG@DA@VEGF in improving angiogenesis and re-epithelialization, while promoting collagen deposition, thus accelerating wound healing with excellent biosafety [316]. In this field, algae-deriving alginate was used to develop SAG wound dressings, which were biodegradable and non-adherent to wound tissues, as well as demonstrated inflammatory effects, which accelerate the healing process significantly. The additional capacity of SAGs to incorporate bioactive substances further improves their effectiveness in complex wound management scenarios [199]. López-Iglesias et al. processed chitosan (CH) in the form of aerogels, thus gifting CH with enhanced water sorption capacity and air permeability. Subsequently, vancomycin was entrapped in its scaffold observing high drug loading capacity and a fast drug release that permitted it to efficiently achieve local therapeutic levels. In vitro studies using fibroblasts and antimicrobial tests against S. aureus showed that the vancomycin-loaded AGs possessed good cytocompatibility and were effective in preventing high bacterial loads at the wound site [315]. Afrashi et al. [317] engineered CH SAGs loaded with vancomycin, observing the same outcomes observed by López-Iglesias et al. [315]. Batista et al. carried out an in vitro study, which evidenced that, when the poor soluble ketoprofen was administered by SAGs, it exhibited release rates improved with respect to its crystalline form, further underling the potential of SAGs to enhance the efficacy of hydrophobic drugs [318]. Collectively, due to the good mucoadhesive properties of CH and its composites, they have appeared as promising matrices to incorporate active pharmaceutical ingredients (API) and other bioactive compounds and preparing CH-SAGs [319]. They have been demonstrated to possess optimal drug loading capacity and in vitro release efficiency for various antibiotics [320,321,322], antifungals [323], anti-inflammatory [324,325,326,327], anticancer [328,329] and corticosteroid drugs [330], as well as insulin [331,332,333], enzymes [334], proteins [335,336] and nucleic acids [337]. Table 7 collects some relevant examples of CH SAGs which have been used to engineer various DDSs.

Moreover, other materials were employed to prepare DDSs based on SAGs. Athamneh et al. prepared SAG microspheres using alginate and hyaluronic acid as ingredients for the drug delivery of pulmonary drug delivery by an emulsion gelation process, followed by supercritical CO_2_ drying. The measured properties by various techniques of the alginate–hyaluronic acid microspheres suggested that they could be suitable as drug carrier for pulmonary delivery [350]. Later, the same authors used AGs technology to develop products like those previously reported loaded with sodium naproxen as pulmonary DDSs [351]. In this case, authors also carried out biological essays missing in the previous study. In vitro drug release investigations showed a non-Fickian drug release mechanism, with no significant difference between the release profiles of alginate and alginate–hyaluronic acid microspheres [351]. In addition to the great contributions that SAGs can provide in drug delivery and wound care, it is expected that the ongoing research in the biomedical application of SAGs will revolutionize traditional therapeutic strategies, thus ameliorating patients’ conditions suffering from various diseases. As an example, Appendix A shows the release mechanism of temperature/pH-responsive carboxymethyl cellulose/poly (*N*-isopropyl acrylamide) interpenetrating polymer network AGs for the delivering of 5-fluorouracile.

#### 4.1.2. Challenges in Practical Applications and Solutions in Using AGs for Building Insulation: Authors Considerations

While AGs boast several advantages and display promising development and application scenarios, several hurdles remain unsolved in the engineering implementation of AGs-based products for insulation. Firstly, an incomplete understanding of thermal insulation mechanisms, nascent supporting systems and inspection methodologies persist, as well as prohibitive costs, which hamper their widespread adoption in the sector of building insulation. Future research in the industrialization technology of AGs is necessary, which should focus on the comprehension of thermal insulation and heat transfer mechanisms at the nanometre scale. They should include the study of the scale, interface and coupling effect of AGs materials, which are still unclear so far. Furthermore, standardized and unified calculation models and testing methods should be developed. The absence of these skills precludes the establishment of national and industry standards for AGs insulation products for construction. Until now, in engineering applications, AGs are mainly promoted through pilot projects and expert demonstrations, while standards and robust groundworks are lacking in applying existing procedures for real energy-saving design. Conventional insulation materials rely predominantly on thermal conductivity values for assessment. Conversely, in cases of AG-based insulation materials, convection and radiation are also to be considered and quantifying the contribution of heat radiation poses a significant challenge. Additionally. the high cost of raw materials to synthesize AGs is another obstacle hindering the widespread adoption of AGs thermal insulation materials. Both supercritical and normal pressure extraction still encounter issues such as high energy consumption, low daily production rates, and protracted production cycles. Unfortunately, there has been no breakthrough in developing a low-cost production process for synthesizing AGs, while the quantity of building insulation material required is huge compared to other industries. In this regard, the price of AGs thermal insulation materials will be pivotal in determining their scalability. Additional efforts should be focused on large-volume, low-cost building AGs, overcoming production process barriers, enhancing production efficiency, and on developing procedures to combine AG raw materials with other materials. Such combinations could minimize the use of AGs to leverage the synergistic advantages of AGs and traditional insulation materials. Interdisciplinary research aimed at thoroughly considering social, economic, environmental and engineering factors to develop cost-effective, low-thermal conductivity, Class A non-combustible materials is mandatory. Primary technical approaches are involving the utilization of inexpensive inorganic silicon sources, such as water glass, silica sol, etc., as raw materials, in place of costly organic silicon raw materials, coupled with supercritical technology and hydrophobic modification techniques to produce low-cost, high-strength AG particles. Currently, research is focusing on developing an inorganic adhesive system using waste fly ash as the matrix. By physically blending the adhesive with AG particles, additives, and aggregates, low-cost, low-thermal conductivity, Class A non-combustible materials—silica AG insulation boards are fabricated through compression moulding and drying processes. Employing insulated decorative integrated board manufacturing technology enables the prefabrication of building components, addressing current challenges in building exterior wall insulation and offering insights for green building energy conservation and insulation.

### 4.2. Metal Oxide-Based Aerogels (MOAGs)

OAGs, also named metal oxide AGs (MOAGs), can be synthesized using almost all transition metal elements of the IV and V group in the periodic table [352]. Anyway, alumina, zirconia and titania, being the ionic crystals with high ionic bond energy and high melting points, can impart the related AGs excellent thermal and chemical steadiness at high temperatures, thus being appropriate for high-temperature applications The most adopted synthetic methods for the preparation of single-component MOAGs include the metal alkoxide sol–gel method if metal alkoxides precursors are used and the EA method if inorganic salts precursors are selected. After gel formation via hydrolysis and condensation, a specific drying process is needed to evade the breakdown of the gel structure. As described in detail in previous sections, the three available drying methods include supercritical fluid (usually ethanol or CO_2_) drying (SCD), ambient pressure drying (APD) and vacuum freezing drying (VFD). In this context, alumina (ALAGs), zirconia (ZRAGs) and titania (TIAGs) AGs can be easily fabricated by these routes, due to their ready availability and controllable precursors. Specific information concerning ALAGs, ZRAGs and TIAGs has been included in Table 8 and Table 9, while Figure 9 schematize the sol–gel process leading to MOAGs [353].

A common approach to control the hydrolysis and condensation reaction rates, thus obtaining an intact gel network, rather than a precipitate, consists of using chelating agents such as acetylacetone (ACAC), ethyl acetoacetate (ETAC), acetic acid and long-chain carboxylic acids [354].

#### 4.2.1. Alumina Aerogels (ALAGs)

Alumina (Al_2_O_3_) is an ionic crystal with a high ionic bond energy and high melting point (2054 °C), mainly used as a high-temperature refractory material, for fabricating firebricks, crucibles, porcelain, and artificial stones. Alumina is available as α-Al_2_O_3_ also indicated as the most thermodynamically stable corundum form or as many metastable η, γ, δ, and θ Al_2_O_3_ forms. Face-centred cubic structures (FCCs) and hexagonal close-packed structures (HCPs) are the most common configuration of alumina according to the arrangement of oxygen ions. As reported by Wu et al. [353], Yoldas and his groups were the first who prepared ALAGs in 1975 using AIP and ASB alumina alkoxides as precursors and chelating agents or complex solvent mixtures to control the hydrolysis and condensation reaction. Unfortunately, the pore structure of alumina aerogels collapses after calcination at high temperature, due to the rapid dehydration and condensation of surface hydroxyl groups and phase transition.

##### Case Studies Concerning ALAGs

Yoldas was the first to report the preparation of alumina oxide (Al_2_O_3_)-based AGs (ALAGs) from alkoxides. The product was not completely in compliance with the porosity requirements for be defined AGs, but they were the first example of highly porous non-siliceous scaffolds [355,356]. The ALAGs were heat treated at 1000 °C, thus achieving transparent discs or rectangles with 4.8 mm thickness. Transparency was improved by impregnating specimens with benzyl alcohol. Several experimentations were therefore finalized to study the intermediate species [357] to improve the mechanical properties [358] and to ameliorate thermal stability [359]. ALAGs are mainly applied as catalysts and catalysts supports [360,361,362,363] for the removal of mustard gas [9] and for capturing cometary fragments, in place of using silica-based compounds [364]. Horiuchi et al. used AIP and ultra-pure water to produce a boehmite sol and, upon addition of nitric acid, ethanol replacement, and supercritical drying ALAGs were reached, which were analyzed to assess their SSA, crystalline phase, particle size with the change in pH, bulk densities, and their relationship with thermal treatment temperature [365]. The sample with a lower bulk density (0.06 g/cm^3^) had a higher θ-α phase transition temperature (1300 °C) and higher SSA (12 m^2^/g at 1300 °C). Pierre et al. synthesized ALAGs in organic solvent using ASB as precursor, and ethyl acetoacetate (ETAC) as chelating agent for slowing down the hydrolysis rate of ASB, whose carbon side groups remained in the structure of ALAG, thus reducing its thermal stability drastically [366]. Transparency was achieved after heat treatment at 900 °C. Kim et al. fabricated alumina aerogel/xerogel via solvothermal reaction methods using AIP precursors and five different alcoholic solvents followed by calcination from 500 to 1200 °C [367]. The authors found that the ALAGs derived from EtOH showed a stable θ-Al_2_O_3_ phase and highest SSA of 66 m^2^/g at 1200 °C. Other authors used inorganic salts in place of alumina oxides to prepare ALAGs via a PO addition method, as a capture agent of H^+^, which is simple, low cost and environmentally friendly method, widely used in the synthesis of other types of metal oxide aerogels. 

Alumina aerogel from AlCl_3_·6H_2_O by PO addition method using PO as a gelation initiator and polyethylene oxide (PEO) as a phase separation inducer, to control the sol–gel reaction have been reported [368]. By proper analytical techniques, authors concluded that the ALAGs had a hierarchical porous structure, and the size of the primary particles and secondary particles were greatly influenced by the drying process and sol–gel parameters [368]. Bao et al. engineered ALAGs using oil shale ash, and several metal oxides, including 26% aluminum oxide, which was leached by sulfuric acid, sodium hydroxide and filtration [369]. The alumina gel formed upon addition of hydrochloric acid and PO. The obtained alumina aerogels after dehydration demonstrated a mesoporous structure with a maximum pore size around 20 nm and flaky morphology. By the incorporation of inorganic ceramic fibres, Yang et al. improved the mechanical properties of ALAGs, achieving a nanocomposite with superior creep resistance, thermal stability and heat insulation properties at 800 °C [370]. Wu et al. fabricated aerogel-like alumina materials using the epoxide addition method replacing the solvent with isopropanol and concluding the synthesis with APD method [371]. The as prepared material demonstrated bulk density (0.133 g/cm^3^) and SSA (505.6 m^2^/g) very close to the values of traditional ALAGs resulting from the SCD method. Collectively, researchers mainly follow two procedures to achieve ALAGs. The first is the alkoxide method completed with SCD, and the second is the inorganic salt method accomplished with APD. Despite, the alkoxide method requires complex chemical reactions to be controlled and risky drying processes, whereas the resultant ALAGs possess better heat resistance up to 1200 °C, due to a better crystallinity and skeleton strength. Conversely, the inorganic salt process, despite simpler, safer and appropriate for large-scale production, provides ALAGs with larger shrinkage, worse crystallinity and thermal stability. However, when compared with other metal oxide aerogels, ALAGs possess the best heat resistance and lowest density and are particularly suitable for high-temperature thermal insulation applications.

#### 4.2.2. Zirconia Aerogels (ZRAGs)

Zirconia is an ionic crystal, with ionic bond energy larger than that of alumina and a higher melting point of 2680 °C, thus having great potential application in high-temperature fields. Zirconia presents three crystal structures, including a monoclinic phase (m-ZrO_2_), tetragonal phase (t-ZrO_2_) and cubic phase (c-ZrO_2_) [372]. Specifically, ZRAGs are produced in the t-ZrO_2_ phase at temperature close to 500 °C and present a surface energy of t-ZrO_2_ lower than that of m-ZrO_2_, a skeleton in the amorphous state, and particle size of about 10 nm. ZRAGs transform partially or completely into m-ZrO_2_ phase when temperature increases to 500 °C, which presents crystals of 30 nm in size [373]. Teichner et al. were the first to prepare ZRAGs in 1976 starting from a zirconium alkoxide as the precursor and completing the synthesis by the SCD method, thus obtaining materials with high SSA, low thermal conductivity, and stable chemical properties [374]. The as-prepared ZRAGs could be excellent candidates for engineering thermal insulators, as well as to develop catalysts and catalyst supports. Unfortunately, ZRAGs tend towards drastic structure collapse and high shrinkage after heat treatments higher than 600 °C, which are attributable to the phase transition and dehydration condensation of hydroxyls, with consequent decreasing of SSA, significantly lower than that of ALAGs. This is the main problem in the field, which limits the ZRAGs high-temperature applications.

##### Case Studies Concerning ZRAGs

By various traditional sol–gel approaches using alkoxides or salts as starting materials followed by calcination, several types of ZRAGs have been produced [11,375,376,377,378,379,380]. Baiker et al. studied the effect of acids and solvents on the structural properties of ZRAGs, while Zeng et al. showed their short-range order and the fractal properties by X-ray diffraction (XRD) and small angle X-ray scattering (SAXS) techniques [381,382,383,384]. ZRAGs are usually applied as catalysts [385,386,387], whose activity was improved by their impregnation with sulphur [388,389,390,391,392,393,394,395], for the phospho-peptide enrichment [396] and thermal insulation [397]. Also, ZRAGs stabilized with Y_2_O_3_ [398,399], doped with Rh398, Ni [392,394], Fe [400], Cu [401] and Co [393], and mixed oxides with Al_2_O_3_ [402], TiO_2_ [403] or WO_x_ [388] were prepared. Jung et al. engineered ZRAGs using different surfactants, such as Brij S10 (C_18_H_37_(OCH_2_CH_2_)_10_OH), triblock copolymer Pluronic P-123 (EO_20_PO_70_EO_20_), and hexadecyltrimethylammonium bromide (CTAB), while N-propanol (n-PrOH) and glacial acetic acid (HAc) were employed as solvent and catalyst, respectively [404]. The authors observed that the different molecular weight and electrostatic attractive force of surfactants affected the pore size, SSA, and crystal growth of ZRAGs, being the cationic CTAB the best enhancer one. By using APD as desiccation method, Brij-76 (C_58_H_118_O_21_) as surfactant, acetic acid as a catalyst and hexamethyldisilane (HMDZ) as silylating agent, Bangi et al. manufactured ZRAGs powders, starting from ZPO as precursors [405]. The presence of Brij-76 decreased the SSA of ZRAGs from 203 to 178 m^2^/g at room temperature (RT) and increased their SSA (27 vs. 45 m^2^/g) at 500 °C. Two years later, the same authors used a simpler method to fabricate zirconia aerogel powders via ZPO as precursors and sulfuric acid (H_2_SO_4_) as a chelating agent, avoiding the use of surfactants via an APD process [9]. Although the authors asserted that a minimal concentration of H_2_SO_4_ provided ZRAGs with high SSA (328 m^2^/g at RT), they omitted to mention their high-temperature properties. Monolithic ZRAGs were achieved by Zhong et al. starting from ZrOCl_2_ as precursor and following a nitric acid-assisted PO addition method [102], fine-tuning the microstructure and SSA of zirconia aerogels by regulating the molar ratio of HNO_3_/ZrOCl_2_ and H_2_O/ZrOCl_2_. The as-prepared ZRAGs exhibited SSA of 454 m^2^/g at RT, a c-ZrO_2_ and t-ZrO_2_ mixed phases before 750 °C, which transformed into m-ZrO_2_ phase after heat treatment at 850 °C. Also, monolithic ZRAGs were engineered by Wang et al. following an organic acid-assisted method in place of the PO addition method [406]. The authors experimented that smaller organic acids, such as acetic acid promoted condensation reactions through covalent or hydrogen bonds, providing monolithic ZRAGs with high SSA of 315 m^2^/g at RT and t-m mixed ZrO_2_ phase at 600 °C. Hydrous ZRAGs (ZrO_x_H_y_) were developed by Long et al. starting from anhydrous ZrCl_4_ via PO addition, followed by CO_2_ SCD process, achieving materials with amorphous structure and abundant hydroxyl [407]. Anyway, increasing calcination temperatures up to 450 °C gave ZrO_x_H_y_ aerogels which crystallized to ZrO_2_ (46% cubic phase and 54% monoclinic phase), while 12% cubic phase and 88% monoclinic phase were observed at 600 °C. Due to the BET SSA being 30 m^2^/g, such materials demonstrated good efficiency in the adsorption and degradation of dimethyl methyl phosphonate. Like ALAGs, we can also find ZRAGs deriving from both alkoxides or inorganic salts, associated with APD or SCD methods, which were extensively investigated to assess their influence on crystal phases and SSAs at low temperatures.

#### 4.2.3. Titania Aerogels (TIAGs)

Titania (TiO_2_) is an ionic crystal possessing a melting point of 1870 °C, which is lower than that of alumina (2054 °C) and zirconia (2680 °C). TiO_2_ may survive in three polymorph crystallographic phases, including the stable rutile (r-TiO_2_) and the two metastable anatase (a-TiO_2_) and brookite (b-TiO_2_) phases. The latter is relatively sporadic, therefore the phase transition from anatase (610 °C) to rutile (915 °C) is the most observed one. TIAGs have high porosity and SSA and is widely used as a photocatalyst for water splitting and photodegradation of organic pollutants, mainly due to their translucency or transparency to visible light and chemical stability. Additionally, titania powders are functional as opacifiers to reduce the radiative heat transport at high temperatures, due to their lower thermal conductivity at high temperatures.

##### Case Studies Concerning TIAGs

Like for ALAGs and ARAGs, TIAGs were prepared for the first time by Teichner et al. in 1976 [374], and nowadays they are usually produced using titanium alkoxide (TIP or TBO) or titanium inorganic salts (TiCl_4_). Material resulting from 2-ethoxyethanol are amorphous with a high SSA of 560 m^2^/g, while that provided by isopropanol show a-TiO_2_ phase and SSA of 150 m^2^/g at RT [408]. TIAGs possess poorer mechanical strength and poorer thermal stability than ALAGS, thus tending to a large decline in SSA for heat treatments above 500 °C, which strongly hamper their thermal insulation and photocatalyst applications. In this context, TIAGs attracted the attention of several researchers [11,409], who usually synthesized them, by dispersing the corresponding titanium ethoxide [410], isopropoxide [411,412,413,414,415,416] or butoxide [417,418] in aqueous alcohol or ketone mixtures, via sol–gel methods followed by calcination. TIAGs are mainly used as catalysts [409,412,414], dye-sensitized solar cells [415] and as filling materials for chromatography columns [413]. To modify the porosity of TIAGs, co-gelling approaches with SiO_2_ were attempted [419,420,421,422,423]. Conversely, doping with Eu [424], Fe [425] and Pt, enhanced the photocatalytic activity [426]. Materials with electron scavenging properties were achieved by introducing gold nanoparticles AuNPs as guests during gelling of TIAGs which acted as host [427]. TIAGs composites with CeO_2_ [428], RuO_2_ [429], MnO*_x_*/V_2_O_5_ [430] and ZrO_2_ [431] with TiO_2_ were prepared to tune the photocatalytic activity. Lermontov et al. synthesized TIAGs using isopropanol, 2-methoxyethanol, and 2-ethoxyethanol as solvents, observing that the material resulting from 2-ethoxyethanol was amorphous with a high SSA of 560 m^2^/g, while that provided by isopropanol showed a-TiO_2_ phase and SSA of 150 m^2^/g at RT [408]. The effect of water on the structural properties of TIAGs achieved using TIP as precursor and HNO_3_ as catalyst was studied by Sadrieyeh et al. [432]. The authors found that lower hydrolysis levels (around 3.75) did not give stable wet gel, while hydrolysis levels of about 7.35 provided the most suitable structural properties with an amorphous phase, SSA of 639 m^2^/g at RT, while an anatase phase with SAA of 157 m^2^/g was observed upon calcination at 450 °C for 2 h. Zhang et al. developed monolithic TIAGs in the form of 3D reticulate TiO_2_ nanofibers displaying an anatase crystalline structure, using an economical and green method, which used bacterial cellulose as a template and preceramic polymer as a precursor, followed by freeze-drying process and calcination at 450 °C [433]. By proper analytical techniques, a low bulk density of 0.04 g/cm^3^, moderate mechanical strength, and SSA of 152 m^2^/g were observed for the achieved materials. Due to the worse thermal stability of TIAGs, than ALAGs and ZRAGs, researchers studied to a major extent their photocatalytic properties, showing that usually the anatase phase exhibits better photocatalytic activity than other titania crystal phases, undergoing a phase transition to rutile ones upon heat treatment exceeding 600 °C. Finding methods to keep the anatase phase and high SSA of TIAGs at high temperature is a key challenge for their photocatalytic applications.

#### 4.2.4. Other Metal Oxide Aerogels (OMOAGs)

Iron oxide (Fe_2_O_3_) [434,435,436,437,438], copper oxide (CuO) [439,440], vanadium oxide (V_2_O_5_) [353,441,442,443], chromium oxide (Cr_2_O_3_) [444], zinc oxide (ZnO) [445,446], manganese oxide (MnO_2_) [447,448,449,450], and mixed metal nickel−iron oxide AGs [91] etc., represent other varieties of MOAGs (OMOAGs). OMOAGs are always prepared via inorganic salts following the PO addition method and exhibit a nanostructure experimenting significant collapse upon calcination at 500 °C, which limits their high-temperature applications.

##### Case Studies Concerning OMOAGs

Yoo et al. achieved Fe_2_O_3_ aerogel (FEAGs) powders using a sol–gel method, starting from Fe(NO_3_)_3_·9H_2_O as precursor and completing the preparation with the APD method [438]. The authors found that powders possessed amorphous nanoporous structure, high SSA of 421 m^2^/g at RT, which transformed magnetite phase upon calcination at 300 °C, and to mixed phases of magnetite and hematite when temperature reached 500 °C, with significant structural collapse and decrease in SSA to 26 m^2^/g after heat treatment at 500 °C. Following a method previously disserted [136], Du et al. engineered monolithic copper oxide aerogel (CUAGs) using cupric chloride (CuCl_2_) as precursor via a dispersed inorganic sol–gel method and polyacrylic acid (PAA) to govern the nucleation and growth in the sol. The as-prepared CUAGs showed SSA of 267 mg/cm^3^, uniform microstructure and monoclinic phase after calcination at 500 °C. Vanadium oxide (V_2_O_5_) is a talented material to act as cathode in lithium (Li) batteries, thanks to its capability to intercalate Li ions. In this regard, several scholars investigated the electrochemical properties of V_2_O_5_-AGs (VAAGs) for energy storage [443,451,452,453,454,455,456,457,458,459,460,461,462,463]. In addition to Li^+^, other AGs hosting ions, such as Na^+^, K^+^, Mg^2+^, Ba^2+^, Al^3+^ and Zn^2+^ were experimented [464,465]. The electrochemical properties of VAAGs were modified by including along with V_2_O_5_ precursor, copper (Cu) or zinc (Zn) [466], by manufacturing nanocomposites with RuO_2_ [467] or with poly pyrrole [468], as well as by hosting Ba_0.25_V_2_O_4_ whiskers, as 1D electron conducting additives [469]. Nitridation of V_2_O_5_provided vanadium oxynitrides, which were tested in sensor devices [470]. The association of TiO_2_ and V_2_O_5_ provided AGs endowed with excellent photocatalytic activity [430,471,472,473,474,475,476,477,478]. It has been reported by Wu et al. [353], that VAAGs were developed by Cui et al. employing H_2_O_2_ and V_2_O_5_ powders as raw materials, achieving materials having SSA high to 395 m^2^/g after heat treatment at 100 °C and SSA of 313 m^2^/g after heat treatment at 350 °C. Also, iron–chromium oxide aerogel (FCRAGs), exhibiting SSA (110 m^2^/g) higher than that of pure Cr_2_O_3_ (27 m^2^/g) or Fe_2_O_3_ (13 m^2^/g), due to the high thermal stability of the Fe-Cr-O*_x_* bond, were manufactured by Khaleel et al. utilizing Cr(NO_3_)_3_·9H_2_O and Fe(NO_3_)_3_·9H_2_O as precursors via a PO addition method [479]. Moreover, Hasanpour et al. reported on the efficient photocatalytic degradation activity of cellulose/ZnO aerogels (C-ZNAGs), vs. methyl orange dye, upon five cycles of reuse [445]. C-ZNAGs demonstrated high stability, hemispherical, cone-like, rice grain-like, and flower-like morphologies and SSA from 44 to 353 m^2^/g, based on the composition [445]. The following Table 10 collects details on these relevant case studies.

Furthermore, Gash et al. proposed a PO addition method to promote gelation by catching H^+^, which demonstrated to be very useful to fabricate high valence state (≥3) metal oxide aerogels like Al_2_O_3_, ZrO_2_, Fe_2_O_3_, and TiO_2_ aerogels [135,362]. Precipitates are usually obtained instead of an opaque gel for low valence state (≤2) metal salts like CoO, ZnO, and CuO. The group of Klabunde et al. synthesized and investigated the formation of magnesium-based AGs (MGAGs) [482,483,484,485], other researchers experimented their gelation by adding glycerol and acetic acid, thus producing large and crack-free MGAGs [486], while recently, Hüsing et al. studied the influence of structure directing agents, such as Pluronic 123 [487]. The possible uses for MGAGs include catalysis or toxic gases remediation [488]. It has been reported that the catalytic activity of MGAGs can be improved by an AuNPs cargo onto the surface [489], by co-gelling with VO*_x_* [490,491] or by producing MGZR mixed oxides [492]. Skapin et al. investigated the effects of organic materials components on the properties of chromium AGs (CRAGs) [493]. Effects of fluorination on CRAGs were later studied [494], and fluorination was then also applied to ALAGs [495,496,497]. While partially fluorination on the surface was possible, deep bulk fluorination led to the complete loss of the AG structure [498]. Also, CRAGs were synthesized by Landau [499,500] and co-workers including α-Cr_2_O_3_, α-CrOOH and other compounds as catalysts supports [501]. Younes et al. finally combined Cr_2_O_3_ with Al_2_O_3_ to catalyze the nitroxidation of toluene to benzonitrile [502]. Much more limited literature was found concerning Fe–Cr–Al mixed oxides [503,504], gold–iron oxides [505], MoO_3_ [506,507,508], WO_3_ [509,510], Li_2_O·B_2_O_2_ [511], GeO_2_ [512], ZnO [513] and V doped ZnO [442,514,515,516], SnO_2_ [517,518,519], Sn–Al oxides [520], Nb_2_O_5_ [521], Pd doped CeO_2_ [522], MnO_2_ [523,524], Ta_2_O_5_ [525] and more complex compositions like MgFe_2_O_4_ [526], BaTiO_3_ [527,528,529], SrTiO_3_ [529], PbTiO_3_ [530], Li_4_Ti_5_O_12_ [531], VOHPO_4_·0.5H_2_O [532], La_2_Mo_2_O_9_ [533] and CuO–Zr*_x_*Ce_1−*x*_O*_y_* [534]. Oxide-based AGs from inorganic salts containing chromium and copper ions (II) were also prepared [23,535], while Bi et al. using DIS prepared amorphous nickel oxide-based AGs with enhanced firmness [440].

## 5. Opportunities, Challenges or Both

The market of AGs, which currently accounts for a valuation of about $300 million, is expected to remarkably growth in the next years. This phenomenon is made possible by the unprecedented advantage which could derive from their extensive application as insulative materials and by their versatile features which make them suitable for a variety of applications. When equated to conventional materials used for insulation, such as fibreglass, foam, or polystyrene, AGs demonstrate very lower thermal conductivities of about 15 mW/mK versus that of 40 mW/mK and 30–40 mW/mK of fibreglass and foam insulators, respectively [536,537]. This characteristic, mainly due to AGs high porosity (99%), makes AGs invaluable contributors to critical sectors, like aerospace and to high efficiency building projects, where they prevent undesirable heat transfer and slash significantly energy costs [538]. Nevertheless, the route to their widespread utilization is still challenging, due to AGs’ fragility, difficult scaling-up, high costs of production, and environmental hazards correlated to their manufacturing processes. AGs’ intrinsic breakability, makes them inclined to impairments under mechanical stress, restraining their practicality in circumstances requiring robust handling [539]. The innate fragility of AGs represents a considerable obstacle to their more extensive use, due to their propensity to break under mechanical tension. This brittleness hampers their employment in sectors where robustness is compulsory. Anyway, several reinforcing agents, including various polymers are under study to develop composite AGs, where the low-density and high porosity typical of traditional AGs are maintained, while mechanical strength is improved. By this approach, AGs are developed with acquired sturdiness and capable to withstand dynamic stress, thus making them suitable for industries such as construction, automotive, and aerospace [540]. The still-not-solved problems of high costs production, due to sophisticated manufacturing processes and expensive chemical precursors, as well as the energy demand of supercritical CO_2_ drying methods, necessary to maintain the porous structure in AGs, hamper their scaling up production and extensive utilization. Innovative drying techniques, such as ambient pressure drying, could improve production costs and reduce energy consumption, thus supporting larger-scale production and increased market accessibility [541]. Additionally, the adoption of different, less costly precursors, such as rice husk ash, silica-rich by-products, such as glasses and other agricultural waste, is a novel approach under exploration to reduce costs. The eco-friendly materials are not only cheaper but also readily available as by-products from other industrial activities, could be of great benefit in decreasing production costs and broadening aerogels’ applicability across various industries [542]. So, the integration of AGs production with industries generating such waste materials, can streamline the supply chain and drastically cut material expenses by repurposing waste streams directly into AGs production, nurturing a circular economy where waste is transformed into new value, thereby diminishing environmental impact [543]. This strategy can contribute to waste reduction, enhance environmental sustainability and provide a use for otherwise discarded resources [544]. The research for less hazardous solvents or solvent-free processes can also dismissing the environmental impact, reduce the emission of volatile organic compounds (VOCs), which contribute to air pollution and potential health risks. This approach can support the global sustainability, increase the commercial and industrial viability of AGs and facilitate their transition from academic experimentation or niche markets to more conventional applications [545]. Efforts to reduce harmful emissions and enhancing the eco-friendliness of AGs production, involve also strategies to capture and recycle solvents or to eliminate harmful solvents entirely are continuously undergone, including the use of supercritical carbon dioxide as a greener alternative to traditional organic solvents [546]. The completely solvent-free synthesis routes, not only restrain VOC emissions but also eliminate the need for high-energy drying processes, thereby further decreasing the energy consumption in AGs production, aligning with broader environmental sustainability goals [547]. Also, 3D printing technologies are under study for AGs synthesis, and are promising for manufacturing complex AGs structures with high precision, possessing properties specifically tailored to diverse industrial applications, thus broadening their commercial appeal and utility [548]. Collectively, multidisciplinary research is urgently needed to develop strategies that could join advancements in materials science, for improving their mechanical robustness and scalability, while limiting manufacturing expenses [549,550]. Additionally, further research is expected to make possible the extension of the most common applications of AGs, such as thermal insulation, to other innovative fields like biomedicine and energy storage, thus confirming the versatile potential of AGs to transfigure diverse industries, and aligning with global goals for environmental sustainability [551,552]. This is particularly impactful in cost-sensitive markets like residential construction, where the high price point of AGs can deter their adoption. The complex and energy-intensive production processes, such as supercritical drying, not only inflate the cost but also restrict large-scale manufacturing capabilities [553]. These strategic innovations pave the way for expanding the AGs’ roles beyond traditional insulation, toward other applications such as filtration, sensors, and energy storage systems [554]. It is noteworthy that AGs application in buildings and vehicles, reduce drastically energy consumption, by their insulating capacities. When incorporated in buildings as insulating materials, AGs can minimize heating and cooling demands, thus reducing both the carbon footprint and operational energy costs. Inserting AGs in vehicle for their insulation, enhance fuel efficiency, minimize heating and cooling calls, thereby lowering greenhouse gas emissions [555]. Application of AGs in thermal management systems as solar panels and wind turbines improves the performance and longevity of these devices [556].

## Data Availability

No new data were created or analyzed in this study. Data sharing is not applicable to this article.

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
