# Peer review of "Aerogels Part 1: A Focus on the Most Patented Ultralight, Highly Porous Inorganic Networks and the Plethora of Their Advanced Applications"

_gels, 2025, doi:10.3390/gels11090718_

Round 1

Reviewer 1 Report

Comments and Suggestions for Authors

The manuscript is generally well-written and the content is clear. However, there are several instances where words have been unnecessarily split with hyphens (e.g., mate-rials, consump-tion, proce-dures). I recommend carefully reviewing the text to remove such formatting issues, as they may distract the reader and reduce readability.

Author Response

The manuscript is generally well-written and the content is clear. However, there are several instances where words have been unnecessarily split with hyphens (e.g., mate-rials, consump-tion, proce-dures). I recommend carefully reviewing the text to remove such formatting issues, as they may distract the reader and reduce readability.

I thank the Reviewer for such notification. Sorry, for the inconvenient. I have carefully checked the copy of submission on my PC and such instances are not present. Something should be happened during the uploading. Anyway, I have searched all issues like those notified by the Reviewer, and they have been corrected. Curiously, they were found all in the Case Studies paragraph in Section 4.4.1, thus confirming that something has gone wrong during submission. Please, see lines 904-1112 (revised paper).

Reviewer 2 Report

Comments and Suggestions for Authors

This review provides first an analysis on AGs according to data found in CAS Content Collection. Then, an AGs’ classification based on the chemical origin of their precursors, as well as the different methods existing to prepare AGs and the current optimization strategies have been discussed. Following, focusing on AGs of inorganic origin, silica and metal oxide-based AGs were reviewed, deeply discussing their properties, specific synthesis and possible used. The classes were chosen based on the evidence that they are the most experimented, patented and marketed AGs. Several related case studied have been reported some of which have been presented in reader-friendly tables and discussed. However, there are many errors and deficiencies in the article, which need to be further modified:

  1. Although the Introduction section provides an overview of the history and applications of aerogels, it lacks a brief summary of current research hotspots and future development trends. It is suggested to add a relevant paragraph in the Introduction section.
  2. Although the classification of aerogels is detailed, there is a lack of in-depth discussion on the unique properties and application scenarios of each type of aerogel. For example, when discussing organic aerogels, the following content can be added: "Organic aerogels, such as phenolic resin aerogels, *******" In addition, it is recommended to add a comparative table of the properties of different types of aerogels in this section, such as density, specific surface area, thermal conductivity, mechanical strength, application fields, etc.
  3. To better highlight the advantages of aerogels, it is necessary to add performance comparisons with other materials when discussing the various properties of aerogels. For example, comparisons of the performance of aerogels with other traditional insulating materials (such as fiberglass, foam plastic, and polystyrene).
  4. Although the manuscript lists many application cases, there is a lack of in-depth analysis of these cases. When discussing the application of aerogels in building insulation, more discussions on the challenges in practical applications and solutions should be added. In addition, more experimental data to support the application cases is also necessary.

Author Response

This review provides first an analysis on AGs according to data found in CAS Content Collection. Then, an AGs’ classification based on the chemical origin of their precursors, as well as the different methods existing to prepare AGs and the current optimization strategies have been discussed. Following, focusing on AGs of inorganic origin, silica and metal oxide-based AGs were reviewed, deeply discussing their properties, specific synthesis and possible used. The classes were chosen based on the evidence that they are the most experimented, patented and marketed AGs. Several related case studied have been reported some of which have been presented in reader-friendly tables and discussed. However, there are many errors and deficiencies in the article, which need to be further modified:

  1. Although the Introduction section provides an overview of the history and applications of aerogels, it lacks a brief summary of current research hotspots and future development trends. It is suggested to add a relevant paragraph in the Introduction section.

I thank a lot the Reviewer for this suggestion and sorry for not having considered these important points before. Current research hotspots and future development trends concerning AGs have been included as a new Section 1.1 in lines 81-111.

  1. Although the classification of aerogels is detailed, there is a lack of in-depth discussion on the unique properties and application scenarios of each type of aerogel. For example, when discussing organic aerogels, the following content can be added: "Organic aerogels, such as phenolic resin aerogels, *******" In addition, it is recommended to add a comparative table of the properties of different types of aerogels in this section, such as density, specific surface area, thermal conductivity, mechanical strength, application fields, etc.

I thank the Reviewer for these suggestions, one of which follows my approach used in constructing this review, that is the use of reader-friendly Tables. The Table asked by the Reviewer has been included in the revised manuscript as new Table 2 (lines 265-268). Concerning the other point raised by the Reviewer, I make kindly note her/him that the required paragraph on phenolic resin aerogels was already present in the original version of the paper. Please, see lines 240-244 (revised manuscript).

  1. To better highlight the advantages of aerogels, it is necessary to add performance comparisons with other materials when discussing the various properties of aerogels. For example, comparisons of the performance of aerogels with other traditional insulating materials (such as fiberglass, foam plastic, and polystyrene).

The required information has been included in lines 185-190 and 201-210.

  1. Although the manuscript lists many application cases, there is a lack of in-depth analysis of these cases. When discussing the application of aerogels in building insulation, more discussions on the challenges in practical applications and solutions should be added. In addition, more experimental data to support the application cases is also necessary.

The required information has been included in the new Section 4.1.2, lines 1138-1182.

The English could be improved to more clearly express the research.

The paper has been revised by Prof Deirdre Kants, our colleague English mothe tongue, who work for University of Genoa and Pavia.

Reviewer 3 Report

Comments and Suggestions for Authors

he paper presents a state-of-the-art overview of aerogels, focusing on their compositions, preparation methods, and the chemical routes involved in their synthesis. In the fourth section, gels are classified according to their chemical composition, offering insight into their applications and practical potential for enhancing modern materials.
The first part of the manuscript highlights the monumental number of papers and patents published over the past 25 years. However, it does not delve into the specific content of these patents. Instead, it categorizes them based on chemical composition and provides an approximate count of publications addressing this topic.
The chemical reactions forming the backbone of the synthesis systems are discussed with the aim of presenting the most common methods for producing aerogels. These reactions are clearly explained and well-supported. The mechanisms are appropriately reported, making this section coherent and informative. I would recommend including a schematic representation of the discussed procedures at the end of Part 3 to enhance clarity and visual comprehension.
The main shortcoming, in my view, is the complete absence of microstructural images illustrating the various structures resulting from different synthesis and drying techniques. This omission significantly limits the reader’s ability to evaluate and compare the structural outcomes of the methods described. Even though the preparation processes are schematically outlined, the lack of actual structural imagery is a major drawback. For instance, the structure of an aerogel synthesized via classical sol-gel methods differs markedly from one derived from nanoparticle precursors. These structural differences can influence the material’s properties and potential applications—points that are mentioned in the text but not visually substantiated.

Page 18. there is something wrong in this sentence they should be there I think 

by modifying the pore sizes, thus enhancing they overall durability, and making them less prone to mechanical failure under stress

same page the phrase needs ending

Pettignano et al. explored alginate, as reinforcing material for SAGs, finding enhanced

page 21 typo correct please

are increasingly being integrated into numerous engineering sec-tors

Aerogel-based glazing systems are recognized for their superior perfor-mance over conventional 

Page 22

text is organized in one paragraph only, but goes from theme to theme without any visual or organizational break thus making it very difficult to read and understand. And it continues two more pages making it very difficult to follow and to understand as topics vary a lot.

Can you schematize the findings to make them more interesting, and give some illustrations please. 

Author Response

The paper presents a state-of-the-art overview of aerogels, focusing on their compositions, preparation methods, and the chemical routes involved in their synthesis. In the fourth section, gels are classified according to their chemical composition, offering insight into their applications and practical potential for enhancing modern materials.
The first part of the manuscript highlights the monumental number of papers and patents published over the past 25 years. However, it does not delve into the specific content of these patents. Instead, it categorizes them based on chemical composition and provides an approximate count of publications addressing this topic.
The chemical reactions forming the backbone of the synthesis systems are discussed with the aim of presenting the most common methods for producing aerogels. These reactions are clearly explained and well-supported. The mechanisms are appropriately reported, making this section coherent and informative. I would recommend including a schematic representation of the discussed procedures at the end of Part 3 to enhance clarity and visual comprehension.

I thank a lot the Reviewer for this suggestion. The required schematic representation of the main synthetic methods described in Section 3 has been added at its end as new Figure 8 (lines 715-717). The numbering of following Figures has been updated.

The main shortcoming, in my view, is the complete absence of microstructural images illustrating the various structures resulting from different synthesis and drying techniques. This omission significantly limits the reader’s ability to evaluate and compare the structural outcomes of the methods described. Even though the preparation processes are schematically outlined, the lack of actual structural imagery is a major drawback. For instance, the structure of an aerogel synthesized via classical sol-gel methods differs markedly from one derived from nanoparticle precursors. These structural differences can influence the material’s properties and potential applications—points that are mentioned in the text but not visually substantiated.

I thank a lot the Reviewer for this suggestion. Required structural images of aerogels prepared using methods described in this paper have been included in the new Supplementary Material file as Figures S1-S9. The description of the images has been included in the main text. Please, see lines 373-384, 470-477, 529-547, 572-588, 596-603, 639-651 and 701-714.

Page 18. there is something wrong in this sentence they should be there I think 

by modifying the pore sizes, thus enhancing they overall durability, and making them less prone to mechanical failure under stress

Sorry for this inconvenient. The sentence has been reformulated (lines 777-780).

same page the phrase needs ending

Pettignano et al. explored alginate, as reinforcing material for SAGs, finding enhanced

Sorry, again. The phrase was completed (lines 805-806).

page 21 typo correct please

are increasingly being integrated into numerous engineering sec-tors

Aerogel-based glazing systems are recognized for their superior perfor-mance over conventional

I thank the Reviewer for having noted the typos. Sorry, for these. Such typos as well as several other similar ones in this Section have been corrected (lines 904-1112, revised paper).

Page 22

text is organized in one paragraph only, but goes from theme to theme without any visual or organizational break thus making it very difficult to read and understand. And it continues two more pages making it very difficult to follow and to understand as topics vary a lot.

Can you schematize the findings to make them more interesting, and give some illustrations please. 

I thank the Reviewer for this suggestion. The findings have been separated according to the main topics reported using sub-headings and some illustration has been provided in Supplementary Materials file as Figure S10-12. The description of the images has been included in the main text. Please, consider lines 904, 966-978, 979, 1009, 1029-1039, 1040, 1068 and 1134-1137.

Reviewer 4 Report

Comments and Suggestions for Authors

Aerogels Part 1:  A Focus on the most patented ultralight, highly porous inorganic networks and the plethora of their advanced applications  (recommended title:  Aerogels:  Review of oxide-based aerogels and their applications)

I do not know what to say or recommend.  I agreed to review because the title and abstract were intriguing.  However, the article is exceedingly long and difficult to read.  What is going to be covered in Part 2?

I do not understand why aerogels are abbreviated to AG.  Why not just call them aerogels?

This review has 556 references.  It is difficult to perform a review of such a large number of references.  It would be more useful to be selective and to choose those references that lead to successful outcomes.  There are so many words that are close to the common usage but not precise.  Beginning with the abstract, the word “nonpareil” should simply be “unusual”.  That is just one example of many ways that the article appears to be written by AI and not a materials scientist.  I recognize that the author is from a Department of Pharmacy.  Trying to translate materials processing in order for a wider audience of scientists to find applications is a good idea.  However, some of the benefits of aerogels in this wider field need to be more carefully described.

Also, the paper gives the impression that aerogels are still a laboratory curiosity.  There are several companies that have manufactured large quantities of aerogel for more than 20 years.  This should be highlighted in the patent literature.

The parts of the article that are most useful are the Tables.  These collections of precursors, applications, and methods show that the author is trying to organize a large quantity of information.  A review focused on the Tables may be easier to appreciate.  This would remove the speculation about usefulness and decreased the excessive description.  In fact, more Tables with physical properties would streamline the review. 

I found some useful information in this review.  I recommend the author engage a copy editor to improve the text. 

Comments on the Quality of English Language

In many cases, 6 words are used when 3 would be enough.  Please edit.

Author Response

Aerogels Part 1:  A Focus on the most patented ultralight, highly porous inorganic networks and the plethora of their advanced applications  (recommended title:  Aerogels:  Review of oxide-based aerogels and their applications)

I thank the Reviewer for this suggestion. Anyway, since a review “Aerogel Part 2” exists and is currently under review in IJMS journal, the use of “Aerogel Part 1”, in my opinion, is mandatory. Additionally, since “Aerogel Part 2” (complete title “Aerogels Part 2. A Focus on the Less Studied Airy Inorganic Networks Despite the Plethora of Possible Advanced Applications”), as the title indicates, deals with chalcogenides and metal-based aerogels (AGs) which, unlike AGs considered in this study, are the less studied and concretely applied AGs, I thought the title of this review in this form could be essential to give continuity to the entire study. For these reasons, I ask kindly the Reviewer to not force me to change the title I chose.

I do not know what to say or recommend.  I agreed to review because the title and abstract were intriguing.  However, the article is exceedingly long and difficult to read.  What is going to be covered in Part 2?

I'm truly delighted that the Reviewer is so curious about the contents of “Aerogel Part 2”, whose title I've already spoiled to satisfy her/his curiosity. However, I'm sorry, but to learn more about its contents in detail, she/he'll have to wait for its publication in IJMS, which I hope will happen as soon as possible.

I do not understand why aerogels are abbreviated to AG.  Why not just call them aerogels?

AG is an abbreviation commonly used for aerogels.

Despite, it could seem that the word “aerogel”, since brief, could have no need of any abbreviation, when associated to other indications concerning the types of AGs intended, abbreviation is of help. So, why not using AG from the beginning?

This review has 556 references.  It is difficult to perform a review of such a large number of references.  It would be more useful to be selective and to choose those references that lead to successful outcomes. 

I understand that the Reviewer could be annoyed with such several references. Sorry for this. Anyway, the numerosity of case studies handled in this paper, need each one its reference, thus rendering necessary several citations. The suggestion of the Reviewer of selecting and citing only the successful outcomes, in my opinion, is limiting. The readers need to be made aware of both positive and negative studies on AGs to gain a holistic view.

There are so many words that are close to the common usage but not precise.  Beginning with the abstract, the word “nonpareil” should simply be “unusual”. 

I'm sorry to have to disagree with the Reviewer, but "nonpareil" does not mean "unusual," It means incomparable, without equal. In fact, I did a little research on synonyms for nonpareil founding several ones, including, having no match or equal; unrivaled, incomparable, matchless, unrivaled, unparalleled, unequaled, without equal, peerless, unmatched, beyond comparison, beyond compare, second to none, unsurpassed, unsurpassable, unbeatable, inimitable, unique and several others exist, but “unusual” is not included between synonyms of nonpareil, as the Reviewer can verify.

That is just one example of many ways that the article appears to be written by AI and not a materials scientist.

This Reviewer's comment may be offensive to me. I'm very sorry the Reviewer felt this way. However, if this is her/his opinion, I can only reassure her/him that it is not, as I stated when submitting the work to Gels (MDPI). I also reassure the Reviewer, that MDPI journals have software to verify whether the papers submitted to them has been done with AI.

I recognize that the author is from a Department of Pharmacy.  Trying to translate materials processing in order for a wider audience of scientists to find applications is a good idea.

I don't fully understand what the Reviewer means with this comment, so I can't be sure to properly respond. I don't understand what she/he means by acknowledging that I'm from the Department of Pharmacy. If she/he intended that for such affiliation I could not be so competent to write a review on aerogels without the aid of AI, I specify that I am an organic chemist, working on nanomaterials of different type and I am the author of two experimental papers on aerogels. Please, consider:

Alfei, S.; Giordani, P.; Zuccari, G. Synthesis and Physicochemical Characterization of Gelatine-Based Biodegradable Aerogel-like Composites as Possible Scaffolds for Regenerative Medicine. Int. J. Mol. Sci. 202425, 5009. https://doi.org/10.3390/ijms25095009.

Alfei, S.; Pintaudi, F.; Zuccari, G. Synthesis and Characterization of Amine and Aldehyde-Containing Copolymers for Enzymatic Crosslinking of Gelatine. Int. J. Mol. Sci. 202425, 2897. https://doi.org/10.3390/ijms25052897.

However, some of the benefits of aerogels in this wider field need to be more carefully described.

In the revision work, several benefits of aerogels previously not reported have been added and several additional images have been included in the new Supplementary Materials file appositely created.

Also, the paper gives the impression that aerogels are still a laboratory curiosity.  There are several companies that have manufactured large quantities of aerogel for more than 20 years.  This should be highlighted in the patent literature.

I am fully aware that some types of aerogels are not only studied at the academic level and are not limited to the laboratory setting, but they are actually used and have been on the market for years. Particularly, SAGs are widely marketed as thermal and acoustic insulators and are widely used in aerospace sector. In this regard, I kindly point out to the Reviewer that SAGs and other AGs are already extensively marketed was already widely described and highlighted in my original paper and even more so in the revised version. I hope this is sufficient for the Reviewer.

The parts of the article that are most useful are the Tables.  These collections of precursors, applications, and methods show that the author is trying to organize a large quantity of information.  A review focused on the Tables may be easier to appreciate.  This would remove the speculation about usefulness and decreased the excessive description.  In fact, more Tables with physical properties would streamline the review. 

I personally love Tables in review articles, and as a Reviewer, I always insist that authors include some explanatory Table (when they're absent) in the reviews. Therefore, I thank the Reviewer very much for this comment, which I assume as positive. 

I found some useful information in this review.  I recommend the author engage a copy editor to improve the text. 

I assure the Reviewer, that the text has been improved.

The English could be improved to more clearly express the research.

The paper has been revised by Prof Deirdre Kants, our colleague English mothe tongue, who work for University of Genoa and Pavia.

Round 2

Reviewer 2 Report

Comments and Suggestions for Authors

All questions have been answered, it can be accepted now.